# TreeGrad-Ranker: Feature Ranking via $O(L)$-Time Gradients for Decision Trees

**Weida Li[1], Yaoliang Yu[2,3], Bryan Kian Hsiang Low[1]**

[1]Department of Computer Science, National University of Singapore, Republic of Singapore
[2]School of Computer Science, University of Waterloo, Canada
[3] Vector Institute, Canada
`vidaslee@gmail.com, yaoliang.yu@uwaterloo.ca,`
`lowkh@comp.nus.edu.sg`

## ABSTRACT

We revisit the use of probabilistic values, which include the well-known Shapley and Banzhaf values, to rank features for explaining the local predicted values of decision trees. The quality of feature rankings is typically assessed with the insertion and deletion metrics. Empirically, we observe that co-optimizing these two metrics is closely related to a joint optimization that selects a subset of features to maximize the local predicted value while minimizing it for the complement. However, we theoretically show that probabilistic values are generally unreliable for solving this joint optimization. Therefore, we explore deriving feature rankings by directly optimizing the joint objective. As the backbone, we propose TreeGrad, which computes the gradients of the multilinear extension of the joint objective in $O(L)$ time for decision trees with $L$ leaves; these gradients include weighted Banzhaf values. Building upon TreeGrad, we introduce TreeGrad-Ranker, which aggregates the gradients while optimizing the joint objective to produce feature rankings, and TreeGrad-Shap, a numerically stable algorithm for computing Beta Shapley values with integral parameters. In particular, the feature scores computed by TreeGrad-Ranker satisfy all the axioms uniquely characterizing probabilistic values, except for linearity, which itself leads to the established unreliability. Empirically, we demonstrate that the numerical error of Linear TreeShap can be up to $10^{15}$ times larger than that of TreeGrad-Shap when computing the Shapley value. As a by-product, we also develop TreeProb, which generalizes Linear TreeShap to support all probabilistic values. In our experiments, TreeGrad-Ranker performs significantly better on both insertion and deletion metrics. Our code is available at `https://github.com/watml/TreeGrad`.

## 1 INTRODUCTION

So far, decision trees remain competitive as recent empirical studies show that gradient-boosted decision trees (Friedman, 2001) tend to perform better on larger or irregular tabular datasets compared to neural networks (Grinsztajn et al., 2022; McElfresh et al., 2024). Often, decision trees are deemed easier to interpret due to their rule-based structures.

In explaining model predictions, the Shapley value (Shapley, 1953) has gained attention due to its uniqueness in satisfying the axioms of linearity, null, symmetry, and efficiency (Lundberg & Lee, 2017). Generally, it is intractable to compute the Shapley value exactly. Thus, many efforts have focused on developing efficient Monte-Carlo methods (e.g., Jia et al., 2019; Covert & Lee, 2021; Zhang et al., 2023; Kolpaczki et al., 2024; Li & Yu, 2024a;b) or frameworks that train models to predict the Shapley value (Jethani et al., 2021; Covert et al., 2022).

Despite that, the Shapley value for explaining decision trees can be accurately computed in polynomial time (Lundberg et al., 2020; Yang, 2021). To our knowledge, the most efficient algorithm is Linear TreeShap (Yu et al., 2022), which runs in $O(LD)$ time while consuming $O(D^2 + L)$ space.

Here, $D$ denotes the depth of the tree, whereas $L$ refers to the number of leaves. Alternatively, Karczmarz et al. (2022) advocated the use of the Banzhaf value (Banzhaf III, 1965) because it is more numerically stable to compute.

On the other hand, Kwon & Zou (2022b) argue that the Shapley value could be mathematically suboptimal in ranking features and other members of Beta Shapley values (Kwon & Zou, 2022a) tend to perform better. Note that all the concepts mentioned belong to the family of probabilistic values uniquely characterized by the axioms of linearity, dummy, monotonicity, and symmetry (Weber, 1988). Empirically, Kwon & Zou (2022a;b); Li & Yu (2023) note that there is no universally the best choice of probabilistic values in ranking features or data. Then, a question arises: how reliable are probabilistic values in ranking features?

The quality of feature rankings are typically measured by the insertion and deletion metrics (Petsiuk et al., 2018; Jethani et al., 2021; Covert et al., 2022). In this work, we start by demonstrating how co-optimizing these two metrics is closely related to a joint optimization that finds a subset of features to maximize the predicted value while its complement minimizes it instead. Using this perspective, we theoretically establish that

- if an attribution method is linear, which include all probabilistic values, then it is no better than random at optimizing the joint objective, as shown in Proposition 1;
- feature rankings induced by probabilistic values can be obtained by substituting a linear surrogate for the joint objective and solving the resulting linearized optimization; operationally, this linearization does not always faithfully represent the original objective.

To circumvent such pitfalls, we explore deriving feature rankings by directly optimizing the joint objective. The focus of our work is on decision trees, and our algorithmic contributions can be summarized as follows:

- As the backbone, we develop TreeGrad (Algorithm 11), which computes the gradients of the multilinear extension of the joint objective in $O(L)$ time and space for decision trees. These gradients include weighted Banzhaf values (Li & Yu, 2023).
- Building on TreeGrad, we introduce TreeGrad-Ranker (Algorithms 2 and 10), which aggregates gradients while optimizing the joint objective to produce a vector of feature scores. This design is motivated by the fact that any symmetric semi-value can be expressed as an expectation over this gradient field (see Appendix A). In particular, as proved in Theorem 1, the resulting feature scores satisfy all the axioms uniquely characterizing probabilistic values, except for linearity, which itself leads to the unreliability established in Proposition 1.
- Although Linear TreeShap is theoretically guaranteed to compute the Shapley value in polynomial time, as shown in Figure 2, it suffers from severe error accumulation. Based on TreeGrad, we therefore develop TreeGrad-Shap (Algorithms 6 and 7), an equally efficient and numerically stable algorithm for computing Beta Shapley values (Kwon & Zou, 2022a) with integral parameters. Empirically, the numerical error of Linear TreeShap can be up to $10^{15}$ times larger than that of TreeGrad-Shap when computing the Shapley value.
- As a by-product, we propose TreeProb (Algorithm 3), which generalizes Linear TreeShap to all probabilistic values. Specifically, the numerical instability in Linear TreeShap stems from the use of an ill-conditioned Vandermonde matrix. We alleviate this issue by replacing it with a well-conditioned matrix. Although this introduces numerical underflow for deep decision trees, the resulting errors are less severe, as shown in Figure 2.

## 2 BACKGROUND

**Notation.** Let $N$ be the number of features and write $[N] := \{1, 2, \ldots, N\}$. Then, $\mathbf{x} \in \mathbb{R}^N$ denotes a specific sample. For convenience, the cardinality of a set $S \subseteq [N]$ is denoted by its lower-case letter $s$. Meanwhile, we write $S \cup i$ and $S \setminus i$ instead of $S \cup \{i\}$ and $S \setminus \{i\}$.

A trained decision tree $f$ is associated with a directed tree $\mathcal{T} := (V, E)$ where $V$ and $E$ refer to all the nodes and edges. $\mathcal{T}$ is rooted because there exists a unique root $r \in V$ such that for every $v \in V$ we can find a path $P_v \subseteq E$ starting from $r$ to $v$. Moreover, each non-root node $v$ has one parent $m_v$, and each non-leaf node $v$ contains two children $a_v$ and $b_v$. For each non-leaf node $v$, it

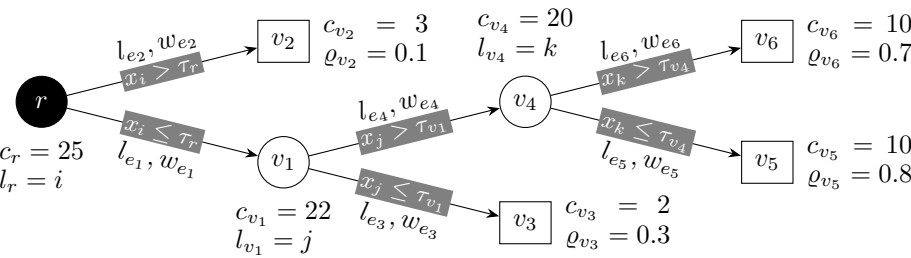

Figure 1: An example of a binary decision tree to illustrate our notation. Each node $v$ is covered by $c_v$ instances. Non-leaf nodes $\textcircled{v}$ split a feature $l_v$ using threshold $\tau_v$ while leaf nodes $\boxed{v}$ make prediction $\varrho_v$. Each edge $e = (u, v)$ is equipped with weight $w_e = c_v/c_u$ and feature $l_e = l_u$.

is associated with a threshold $\tau_v \in \mathbb{R}$ concerning a specific feature $l_v \in [N]$; $\mathbf{x}$ is directed into $a_v$ if $x_{l_v} \leq \tau_v$ and $b_v$ otherwise. For each $v \in V$, let $c_v$ be the number of training data covered by the subtree rooted at $v$. Each edge $e := (u \to v) \in E$ is weighted by $0 < \frac{c_v}{c_u} < 1$, and we write $h_e := v$; plus, $e$ is labeled with $l_u$. Therefore, $w_e$ and $l_e$ refer to the weight and the label of $e$. Then, we define $P_{i,v} := \{e \in P_v \mid l_e = i\}$ and $F_v := \{l_e : e \in P_v\}$. For every edge $e \in P_v$, denote $e^\uparrow$ by its nearest ancestor along $P_v$ that shares the same label with $e$; $e^\uparrow$ may not exist. For each node $v$, let $L_v$ be the set that contains all the leaves of $\mathcal{T}$ that can be reached from $v$. Then, we define $L_v^\dagger := \{v' \in L_v \mid (m_v \to v) \text{ is the nearest edge ancestor of } v' \text{ labeled by } l_{m_v}\}$. For each leaf node $v \in L_r$, let $\varrho_v \in \mathbb{R}$ denote its predicted value. We denote by $D$ the depth of $\mathcal{T}$ and by $L$ the number of leaves of $\mathcal{T}$. Note that $|V| = 2L - 1$ and $D < L$. We assume that all the features in $[N]$ are used by $\mathcal{T}$, and thus $N < |V|$. A simple example with $D = 3$ and $L = 4$ can be found in Figure 1.

For a polynomial $p(y)$ of degree $\deg(p)$, we write $p(y) = \sum_{k=0}^{\deg(p)} p_{k+1} y^k$, i.e., we denote its coefficients by the corresponding bold letter $\mathbf{p} \in \mathbb{R}^{\deg(p)+1}$. Then, the inner product of two polynomials is $\langle p(y), q(y) \rangle := \langle \mathbf{p}, \mathbf{q} \rangle$.

## 2.1 Probabilistic Values and Semi-Values

One popular way of ranking features is the use of probabilistic values, as it uniquely satisfies the axioms of linearity, dummy, monotonicity, and symmetry (Weber, 1988). Precisely, each probabilistic value can be expressed as

$$\phi_i^{\boldsymbol{\omega}}(f_{\mathbf{x}}) := \sum_{S \subseteq [N] \setminus i} \omega_{s+1} \cdot [f_{\mathbf{x}}(S \cup i) - f_{\mathbf{x}}(S)], \tag{1}$$

where $\boldsymbol{\omega} \in \mathbb{R}^N$ is non-negative with $\sum_{k=1}^N \binom{N-1}{k-1} \omega_k = 1$ and $f_{\mathbf{x}}(S) := \mathbb{E}_{\mathbf{X}_{S^c}}[f(\mathbf{x}_S, \mathbf{X}_{S^c})]$ where $\mathbf{x}_S$ is the restriction of $\mathbf{x}$ on $S$ and $S^c := [N] \setminus S$. Following Lundberg et al. (2020), there are two choices for sampling $\mathbf{X}_{S^c}$. One choice is to use the marginal distribution; however, this has been argued to be unreliable, as it uses off-manifold data and thus may produce manipulable explanations (Slack et al., 2020; Frye et al., 2021). We follow the other choice (Algorithm 8), which may be seen as a convenient approximation of the conditional distribution. This approach has recently been adopted for computing Shapley interactions (Muschalik et al., 2024). If we further add the consistency axiom, the family of probabilistic values shrinks, and the remaining members are referred to as semi-values (Dubey et al., 1981; Li & Yu, 2023). Specifically, $\phi^{\boldsymbol{\omega}}$ is a semi-value if and only if there exists a Borel probability measure $\mu$ over the closed interval $[0, 1]$ such that for every $k \in [N]$,

$$\omega_k = \int_0^1 t^{k-1} (1-t)^{N-k} \mathrm{d}\mu(t). \tag{2}$$

In particular, weighted Banzhaf values and Beta Shapley values are semi-values, with the latter including the Shapley value (Shapley, 1953) as a special case. We refer the reader to Appendix A for more details. $\phi^\mu(f_{\mathbf{x}})$ will be used to denote semi-values.

## 2.2 LINEAR TREESHAP

Previously, Yu et al. (2022) introduced Linear TreeShap capable of computing the Shapley value $\phi^{\text{Shap}}(f_{\mathbf{x}})$ in $O(LD)$ time and $O(D^2 + L)$ space. We notice that a counterpart with the same time and space complexities also appeared in Karczmarz et al. (2022, Appendix E), conveniently referred to as TreeShap-K. Since Yu et al. (2022) offer a more compact set of definitions for describing how the relevant terms traverse the tree, we follow the concepts used by Linear TreeShap. In Appendix D, we demonstrate how to evolve Linear TreeShap into TreeShap-K.

**Linearization of the approximation of $f_S$.** For each $i \in [N]$ and $v \in L_r$, we define $x \in \Pi_{i,v}$ if $\mathbf{x}$ satisfies all the splitting criteria along the path $P_v$ that involves feature $i$. The convention is $x \in \Pi_{i,v}$ if $P_{i,v} = \emptyset$. Then, for each $i \in [N]$ and $v \in L_r$, a function $\gamma_{i,v} \colon \mathbb{R}^N \to \mathbb{R}$ is defined by $\gamma_{i,v}(\mathbf{x}) := \mathbb{1}_{\mathbf{x} \in \Pi_{i,v}} \cdot \prod_{e \in P_{i,v}} \frac{1}{w_e}$. Then, $f_{\mathbf{x}}(S) = \sum_{v \in L_r} R_{\mathbf{x}}^v(S)$ where each path-dependent rule function $R_{\mathbf{x}}^v \colon 2^{[N]} \to \mathbb{R}$ is defined by setting $R_{\mathbf{x}}^v(S) := \varrho_v \cdot \frac{c_v}{c_r} \prod_{i \in S} \gamma_{i,v}(\mathbf{x})$.

**The Shapley value of $R_{\mathbf{x}}^v$.** From now on, we will omit $\mathbf{x}$ occasionally for simplicity. Since $\phi_i^{\text{Shap}}(f_{\mathbf{x}}) = \sum_{v \in L_r} \phi_i^{\text{Shap}}(R_{\mathbf{x}}^v)$, it suffices to demonstrate how to compute the Shapley value of each $R_{\mathbf{x}}^v$. According to Yu et al. (2022, Lemma 2.2),

$$\phi_i^{\text{Shap}}(R_{\mathbf{x}}^v) = \frac{\varrho_v c_v (\gamma_{i,v} - 1)}{c_r |F_v|} \sum_{S \subseteq F_v \setminus i} \binom{|F_v| - 1}{s}^{-1} \prod_{j \in S} \gamma_{j,v} = (\gamma_{i,v} - 1) \psi\left( \frac{G_v(y)}{1 + \gamma_{i,v} \cdot y} \right)$$

where $G_v(y) = \frac{\varrho_v c_v}{c_r} \prod_{j \in F_v} (1 + \gamma_{j,v} \cdot y)$ is a polynomial of $y$,[1] and $\psi(p(y)) = \langle p(y), B_{\deg(p)}(y) \rangle$ where $B_d(y) := \frac{1}{d+1} \sum_{k=0}^{d} \binom{d}{k}^{-1} y^k$. Since $\gamma_{i,v} - 1 = 0$ if $i \notin F_v$, the focus is on the cases where $G_v(y)$ is divisible by $1 + \gamma_{i,v} \cdot y$. The key idea is that the coefficient of $y^k$ in $\prod_{j \in F_v \setminus i} (1 + \gamma_{j,v} \cdot y)$ is exactly $\sum_{S \subseteq F_v \setminus i}^{|S|=k} \prod_{j \in S} \gamma_{j,v}$, which shows that the Shapley value of interest can be computed using polynomial operations. Before proceeding, for each edge $e \in E$, define a function $\gamma_e \colon \mathbb{R}^N \to \mathbb{R}$ by letting $\gamma_e(\mathbf{x}) := \mathbb{1}_{\mathbf{x} \in \Pi_e} \cdot \prod_{e' \in P_{l_e, h_e}} \frac{1}{w_{e'}}$ where $\mathbf{x} \in \Pi_e$ if $\mathbf{x}$ satisfies all the criteria along the path $P_{h_e}$ that involves feature $l_e$. Conveniently, if $e$ is the last edge in $P_{i,v}$, then $\gamma_{i,v} = \gamma_e$. While traversing down a path $P_v$, $G_v(y)$ is what Linear TreeShap computes. For traversing up the tree, we adopt the computation proposed by Karczmarz et al. (2022, Lemma 5) as it is more elegant; precisely, $\phi_i^{\text{Shap}}(f_{\mathbf{x}}) = \sum_{\substack{e \in E \\ l_e = i}} \sum_{v \in L_{h_e}^{\dagger}} (\gamma_e - 1) \cdot \psi\left( \frac{G_v(y)}{1 + \gamma_e \cdot y} \right)$.

**Why does Linear TreeShap run in $O(LD)$ time.** The computational efficiency comes from the so-called scaling invariability equality (Yu et al., 2022, Proposition 2.1),

$$\psi(p(y)) = \psi\left( p(y) \cdot (1 + y)^k \right) \quad \text{where} \quad k \geq 0. \tag{3}$$

Accordingly, $\phi_i^{\text{Shap}}(f_{\mathbf{x}}) = \sum_{e \in E, l_e = i} (\gamma_e - 1) \psi\left( \sum_{v \in L_{h_e}^{\dagger}} \frac{G_v(y)}{1 + \gamma_e \cdot y} \cdot (1 + y)^{d_e - \deg(G_v)} \right)$ where $d_e = \max_{v \in L_{h_e}} \deg(G_v(y))$. Clearly, the significance of Eq. (3) is that it is now possible to sum over $\left\{ \frac{G_v(y)}{1 + \gamma_e \cdot y} \right\}_{v \in L_{h_e}^{\dagger}}$ and then compute $\sum_{v \in L_{h_e}^{\dagger}} \phi_{l_e}^{\text{Shap}}(R_{\mathbf{x}}^v)$ using $\psi$ only once. Without Eq. (3), $\psi$ would be evaluated $|L_{h_e}^{\dagger}|$ times instead. Please see Appendix B on how $\psi$ is implemented.

## 3 HOW UNRELIABLE ARE PROBABILISTIC VALUES IN RANKING FEATURES?

Suppose $f$ outputs the probability of a class, and let $\phi \in \mathbb{R}^N$ denotes feature scores for the prediction $f(\mathbf{x})$. Then, a higher $\phi_i$ indicates that the $i$-th feature contributes more positively to increasing $f_{\mathbf{x}}$. Typically, the quality of $\phi$ is measured by the insertion and deletion metrics (Petsiuk et al., 2018). Notice that these two metrics only employ the ranking $\pi : [N] \to [N]$ induced by $\phi$, i.e., $\phi_{\pi(1)} \geq \phi_{\pi(2)} \geq \cdots \geq \phi_{\pi(N)}$. The two metrics are as follows:

$$\text{Ins}(\pi; f_{\mathbf{x}}) := \frac{1}{N} \sum_{k=1}^{N} f_{\mathbf{x}}(S_k^+(\pi)) \quad \text{and} \quad \text{Del}(\pi; f_{\mathbf{x}}) := \frac{1}{N} \sum_{k=1}^{N} f_{\mathbf{x}}(S_k^-(\pi)) \tag{4}$$

---

[1] Since $\omega_\ell = \omega_{n+1-\ell}$ for the Shapley value, $1 + \gamma_{k,v} \cdot y$ is interchangeable with $\gamma_{k,v} + y$.

where $S_k^+(\pi) = \{\pi(1), \pi(2), \ldots, \pi(k)\}$ and $S_k^-(\pi) = \{\pi(N), \pi(N-1), \ldots, \pi(N-k+1)\}$. A higher $\text{Ins}(\pi; f_{\mathbf{x}})$ indicates that the top-ranked features contribute more positively to increasing $f(\mathbf{x})$, whereas a lowers $\text{Del}(\pi; f_{\mathbf{x}})$ indicates that the bottom-ranked features contribute more negatively. Let $S_*^+(\pi) := \arg\max_{S_k^+(\pi)} f_{\mathbf{x}}(S_k^+(\pi))$ and $S_*^-(\pi) := \arg\min_{S_k^-(\pi)} f_{\mathbf{x}}(S_k^-(\pi))$. Then, $S_*^+(\pi)$ can be viewed as the positive features identified by $\pi$, whereas $S_*^-(\pi)$ contains negative ones. Empirically, we observe that a higher $\text{Ins}(\pi; f_{\mathbf{x}})$ is often associated with a higher $f_{\mathbf{x}}(S_*^+(\pi))$, which essentially corresponds to finding a subset $S^+$ that maximizes $f_{\mathbf{x}}(S^+)$. Similarly, the deletion metric is closely related to finding a subset $S^-$ that minimizes $f_{\mathbf{x}}(S^-)$. Meanwhile, it is expected that $S^+ = [N] \setminus S^-$. Therefore, equally co-optimizing $\text{Ins}(\cdot; f_{\mathbf{x}})$ and $\text{Del}(\cdot; f_{\mathbf{x}})$ is closely related to

$$\underset{S \subseteq [N]}{\text{maximize}} \; \frac{1}{2} \left( f_{\mathbf{x}}(S) - f_{\mathbf{x}}([N] \setminus S) \right). \tag{5}$$

Next, we will theoretically demonstrate why probabilistic values are generally unreliable in solving the problem (5).

Consider an arbitrary set function $U : 2^{[N]} \to \mathbb{R}$ and define $D_U$ by letting $D_U(S) = \frac{1}{2}(U(S) - U([N] \setminus S))$. We will show that for each probabilistic value $\phi^{\boldsymbol{\omega}}(U)$, there exist two distinct sets $S_1$ and $S_2$ such that $\phi^{\boldsymbol{\omega}}(U)$ can not reliably differentiate between the two hypotheses $D_U(S_1) \leq D_U(S_2)$ and $D_U(S_1) > D_U(S_2)$. Following Bilodeau et al. (2024); Wang et al. (2024), a hypothesis test is a function $h : \mathbb{R}^N \to [0, 1]$ that takes as input a contribution vector, such as $\phi^{\boldsymbol{\omega}}(U)$, and outputs the possibility to reject the null hypothesis $D_U(S_1) \leq D_U(S_2)$. Then, the reliability of a test $h$ can be bounded.

**Proposition 1.** *Suppose $\phi(f_{\mathbf{x}}) \in \mathbb{R}^N$ is linear in $f_{\mathbf{x}}$ with $N > 3$, which includes all probabilistic values. Then, there exist distinct sets $S_1$ and $S_2$ such that $S_1 \neq [N] \setminus S_2$ and*

$$\text{Rel}(h; S_1, S_2) := \inf_{U : D_U(S_1) \leq D_U(S_2)} [1 - h(\phi(U))] + \inf_{U : D_U(S_1) > D_U(S_2)} h(\phi(U)) \leq 1.$$

In particular, for a random guess $\overline{h} \equiv 0.5$, we have $\text{Rel}(\overline{h}; S_1, S_2) = 1$. Therefore, this proposition indicates that for each probabilistic value, there are hypotheses such that it can not reliably reject. The intuition behind this proposition is that a linear operator maps vectors of dimension $2^N$ into significantly compressed ones of dimension $N$, leaving vast room for not detecting non-negligible differences. This may limit how well probabilistic values are in solving the problem (5), or equivalently, ranking features.

**Proposition 2.** *Let $\pi$ be a ranking induced by a probabilistic value $\phi^{\boldsymbol{\omega}}(f_{\mathbf{x}})$ such that $\phi^{\boldsymbol{\omega}}_{\pi(1)}(f_{\mathbf{x}}) \geq \phi^{\boldsymbol{\omega}}_{\pi(2)}(f_{\mathbf{x}}) \geq \cdots \geq \phi^{\boldsymbol{\omega}}_{\pi(N)}(f_{\mathbf{x}})$. For every $1 \leq k \leq N$, $S_k^+ := \{\pi(1), \pi(2), \ldots, \pi(k)\}$ is an optimal solution to the problem*

$$\underset{S \subseteq [N], |S| = k}{\text{maximize}} \; \frac{1}{2} \left( g^*(S) - g^*([N] \setminus S) \right)$$

*where $g^*$ is the best linear surrogate for $f_{\mathbf{x}}$, i.e., $g^* = \arg\min_{g \in \mathcal{L}} \sum_{S \subseteq [N]} \eta_{s+1} (f_{\mathbf{x}}(S) - g(S))^2$ with $\mathcal{L} := \{g : g(S) = t_0 + \sum_{i \in S} t_i\}$. The weights satisfy (i) $\eta_1, \eta_{N+1} \geq 0$ and (ii) $\eta_s = \omega_s + \omega_{s-1}$ for $2 \leq s \leq N$.*

Let $S_k^* := \arg\max_{S \subseteq [N], |S| = k} \frac{1}{2}(f_{\mathbf{x}}(S) - f_{\mathbf{x}}([N] \setminus S))$. Then, $S_k^+$ can be treated as an approximate solution to $S_k^*$. Taking it a step further,

$$\underset{S \subseteq [N]}{\text{maximize}} \; df_{\mathbf{x}}(S) = \underset{W \in \{S_k^* : 1 \leq k \leq N\}}{\text{maximize}} df_{\mathbf{x}}(W) \approx \underset{W \in \{S_k^+ : 1 \leq k \leq N\}}{\text{maximize}} df_{\mathbf{x}}(W).$$

where $df_{\mathbf{x}}(S) := \frac{1}{2}(f_{\mathbf{x}}(S) - f_{\mathbf{x}}([N] \setminus S))$. It suggests that the use of probabilistic values is equivalent to substituting linear surrogates for $f_{\mathbf{x}}$ while optimizing the problem (5).

**Why not optimize the problem (5) directly?** The problem $\text{maximize}_S \, U(S)$ is known to be NP-hard (Feige et al., 2011). Alternatively, one may resort to a linear surrogate as it is more tractable. However, such a linearization comes at the expense of not accurately representing the behavior of $f_{\mathbf{x}}$. In what follows, we will demonstrate that decision trees are well-structured such that it is practical to solve the problem (5) using gradient ascent. Meanwhile, we are able to obtain feature scores that satisfy all the axioms uniquely characterizing probabilistic values, except linearity.

---

**Algorithm 1** TreeGrad (Simplified Procedure)

---

**Input:** Decision tree $\mathcal{T} = (V, E)$ with root $r$, instance $\mathbf{x} \in \mathbb{R}^N$, point $\mathbf{z} \in [0, 1]^N$
**Output:** Gradient $\mathbf{g}$
1 **Function** `traverse`$(v, s = 1)$**:**
2     $e \leftarrow (m_v \to v)$
3     **if** $e^\uparrow$ *exists* **then**
4       $s \leftarrow \frac{s}{1 - z_{l_{e^\uparrow}} + z_{l_{e^\uparrow}} \gamma_{e^\uparrow}} \cdot (1 - z_{l_e} + z_{l_e} \gamma_e)$
5     **else**
6       $s \leftarrow s \cdot (1 - z_{l_e} + z_{l_e} \gamma_e)$
7     **if** $v$ *is a leaf* **then**
8       $s \leftarrow \frac{\varrho_v c_v}{c_r} \cdot s$
9     **else**
10      $s \leftarrow$ `traverse`$(a_v, s) +$ `traverse`$(b_v, s)$
11    **if** $e^\uparrow$ *exists* **then**
12      $H[h_{e^\uparrow}] \leftarrow H[h_{e^\uparrow}] - s$
13    $H[v] \leftarrow H[v] + s$
14    $g_{l_e} \leftarrow g_{l_e} + (\gamma_e - 1) \cdot \frac{H[v]}{1 - z_{l_e} + z_{l_e} \gamma_e}$
15    **return** $s$
16 Initialize $H[v] = 0$ for every $v \in V$
17 Initialize $\mathbf{g} = \mathbf{0}_N$
18 `traverse`$(a_r)$
19 `traverse`$(b_r)$
20 **return g**

---

**Algorithm 2** TreeGrad-Ranker with Gradient Ascent (GA)

---

**Input:** Decision tree $\mathcal{T} = (V, E)$ with root $r$, instance $\mathbf{x} \in \mathbb{R}^N$, learning rate $\epsilon > 0$, total number of iterations $T$
**Output:** A vector of feature scores $\boldsymbol{\zeta}$
21 Initialize $\mathbf{z} = 0.5 \cdot \mathbf{1}_N$ and $\boldsymbol{\zeta} = \mathbf{0}_N$
22 **for** $t = 1, 2, \ldots, T$ **do**
23    $\mathbf{g} \leftarrow \frac{1}{2}(\text{TreeGrad}(\mathcal{T}, \mathbf{x}, \mathbf{z}) + \text{TreeGrad}(\mathcal{T}, \mathbf{x}, \mathbf{1}_N - \mathbf{z}))$
24    $\mathbf{z} \leftarrow \mathbf{z} + \epsilon \cdot \mathbf{g}$
25    $\mathbf{z} \leftarrow \text{Proj}(\mathbf{z})$                  `//` $z_i \leftarrow \min(\max(0, z_i), 1)$
26    $\boldsymbol{\zeta} \leftarrow \frac{t-1}{t} \cdot \boldsymbol{\zeta} + \frac{1}{t} \cdot \mathbf{g}$
27 **return** $\boldsymbol{\zeta}$

---

## 4 TREEGRAD-RANKER

To make $f_\mathbf{x} : 2^{[N]} \to \mathbb{R}$ differentiable, the first step is to make its domain continuous. Following Owen (1972), the multilinear extension of $f_\mathbf{x}$, denoted by $\overline{f}_\mathbf{x} : [0, 1]^N \to \mathbb{R}$, is defined by letting $\overline{f}_\mathbf{x}(\mathbf{z}) = \sum_{S \subseteq [N]} \left( \prod_{j \in S} z_j \right) \left( \prod_{j \notin S} (1 - z_j) \right) f_\mathbf{x}(S)$. Then, the problem (5) is relaxed as

$$\underset{\mathbf{z} \in [0,1]^N}{\text{maximize}} \frac{1}{2}(\overline{f}_\mathbf{x}(\mathbf{z}) - \overline{f}_\mathbf{x}(\mathbf{1}_N - \mathbf{z})). \tag{6}$$

This relaxation is justified by the following proposition.

**Proposition 3.** *Suppose the optimal set $S^*$ for the problem (5) is unique, then $\mathbf{1}_{S^*}$ whose $i$-th entry is 1 if $i \in S^*$ and 0 otherwise is the unique optimal solution to the relaxed problem (6).*

Then, according to Owen (1972, Eq. (8)), we have

$$\nabla \overline{f}_\mathbf{x}(\mathbf{z}) = \sum_{S \subseteq [N] \setminus i} \left( \prod_{j \in S} z_j \right) \left( \prod_{j \notin S \cup i} (1 - z_j) \right) [f_\mathbf{x}(S \cup i) - f_\mathbf{x}(S)]. \tag{7}$$

**An $O(L)$-time TreeGrad for computing gradients.** Since $\nabla \overline{f}_{\mathbf{x}}(\mathbf{z}) = \sum_{v \in L_r} \nabla \overline{R}_{\mathbf{x}}^v(\mathbf{z})$, it suffices to simplify the formula of each $\nabla \overline{R}_{\mathbf{x}}^v(\mathbf{z})$.

**Lemma 1.** *We have* $\nabla_i \overline{R}_{\mathbf{x}}^v(\mathbf{z}) = (\gamma_{i,v} - 1) \cdot \frac{H_v}{1 - z_i + z_i \gamma_{i,v}}$, *where* $H_v = \frac{\varrho_v c_v}{c_r} \cdot \prod_{j \in F_v} (1 - z_j + z_j \gamma_{j,v})$.

Starting from $r$, write $P_v = (e_1, e_2, \ldots, e_{|P_v|})$ in the order encountered when traversing down the path $P_v$ to the leaf node $v$. Then, running

$$
s_0 \leftarrow 1 \text{ and } s_\ell \leftarrow
\begin{cases}
\frac{s_{\ell-1}}{1 - z_{k_\ell} + z_{k_\ell} \gamma_{e_\ell^\uparrow}} (1 - z_{j_\ell} + z_{j_\ell} \gamma_{e_\ell}), & e_\ell^\uparrow \text{ exists,} \\
(1 - z_{j_\ell} + z_{j_\ell} \gamma_{e_\ell}) \cdot s_{\ell-1}, & \text{otherwise,}
\end{cases}
$$

where $j_\ell := l_{e_\ell}$ and $k_\ell := l_{e_\ell^\uparrow}$ until it reaches the leaf node $v$, we have $\frac{\varrho_v c_v}{c_r} \cdot s_{|P_v|} = H_v$. While traversing up the tree from leaf nodes, each $\nabla_i \overline{f}_{\mathbf{x}}(\mathbf{z})$ is computed using the following lemma.

**Lemma 2.** $\nabla_i \overline{f}_{\mathbf{x}}(\mathbf{z}) = \sum_{\substack{e \in E \\ l_e = i}} (\gamma_e - 1) \cdot \frac{\sum_{v \in L_{h_e}^\dagger} H_v}{1 - z_i + z_i \gamma_e}$.

As shown in Algorithm 1, lines 2–10 perform the traversing-down procedure, whereas the traversing-up procedure is efficiently carried out in lines 11–15.[2] In line 14, it may occur that $1 - z_{l_e} + z_{l_e} \gamma_e = 0$ when $\gamma_e = 0$ and $z_{l_e} = 1$, which makes the division invalid. This issue can be easily addressed by noting that at most one feature receives a non-zero update in $\mathbf{g}$. The full procedure of TreeGrad that accounts for this corner case is presented in Algorithm 11. Overall, the total running time is $O(L)$ while using $O(L)$ space. Equipped with TreeGrad, we can solve the relaxed problem (6) using gradient ascent, which is shown in Algorithm 2. Specifically, $\epsilon$ is a learning rate and Proj projects each $\mathbf{z}_{t-1}$ onto $[0,1]^N$, i.e., $z_i \leftarrow \min(\max(0, z_i), 1)$. We comment that $\zeta = \nabla \overline{f}_{\mathbf{x}}(0.5 \cdot \mathbf{1}_N)$ when $T = 1$, which is exactly the Banzhaf value. Since any symmetric semi-value, which includes the Shapley value and the Banzhaf value, can be expressed as an expectation of this gradient field (see Appendix A), we take the averaged gradient as the vector of feature scores. Moreover, the gradient ascent used in Algorithm 2 can be replaced by the ADAM optimizer (Kingma & Ba, 2015), resulting in Algorithm 10. In particular, both algorithms verify all the axioms that uniquely characterize probabilistic values except for the linearity axiom.

**Theorem 1.** *The vector of feature scores $\zeta$ returned by TreeGrad-Ranker, either using Algorithm 2 or 10, can be expressed as $\zeta_i = \sum_{S \subseteq [N] \setminus i} \omega_i(S; \mathbf{x})(f_{\mathbf{x}}(S \cup i) - f_{\mathbf{x}}(S))$ where $\omega_i(S; \mathbf{x}) > 0$ and $\sum_{S \subseteq [N] \setminus i} \omega_i(S; \mathbf{x}) = 1$. Moreover, it verifies the axioms of dummy, equal treatment and monotonicity.*

## 5   AN $O(LD)$-TIME TREEPROB FOR COMPUTING PROBABILISTIC VALUES

**Generalize the scaling invariability equality.** To compute probabilistic values, the only difference lies in how $\psi$ is defined. Therefore, to generalize Linear TreeShap for any probabilistic value $\phi^{\boldsymbol{\omega}}(f_{\mathbf{x}})$, it suffices to generalize the scaling invariability equality in Eq. (3). This is straightforward by noticing that multiplying by $(1 + y)$ simply introduces a null player.

**Lemma 3** (Generalization of Eq. (3)). *Suppose $p(y)$ is a polynomial of degree $N - 2$, we have* $\langle p(y) \cdot (1 + y), \sum_{k=0}^{N-1} \omega_{k+1} y^k \rangle = \langle p(y), \sum_{k=0}^{N-2} (\omega_{k+1} + \omega_{k+2}) y^k \rangle$.

Then, $\psi\left(p(y) \cdot (1 + y)^k\right) = \psi\left(p(y)\right)$ in Eq. (3) is simply the result of recursively applying the equality in Lemma 3 while substituting $\omega_\ell = \frac{(\ell-1)!(N-\ell)!}{N!}$. For semi-values, using Eq. (2), their scaling invariability equality can be simplified as

$$
\langle p(y) \cdot (1 + y)^k, q_{\deg(p)+k}^\mu(y) \rangle = \langle p(y), q_{\deg(p)}^\mu(y) \rangle, \tag{8}
$$

where $q_\ell^\mu(y) = \sum_{k=0}^\ell \left( \int_0^1 t^k (1 - t)^{\ell-k} \mathrm{d}\mu(t) \right) y^k$.

---

[2] The original implementation of the traversing-up procedure, as in (Karczmarz et al., 2022, Algorithms 2–4), is unnecessarily complicated; they compute $\nabla \overline{f}_{\mathbf{x}}(0.5 \cdot \mathbf{1}_N)$, equivalently the Banzhaf value.

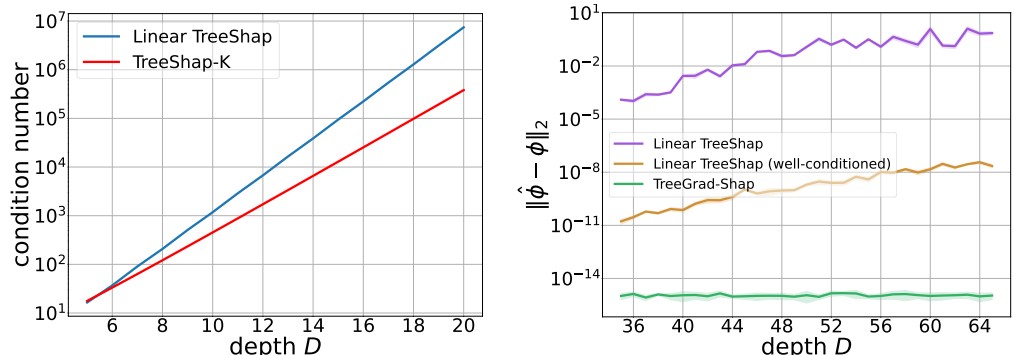

Figure 2: The condition numbers of matrices involved in Linear TreeShap and TreeShap-K, and the inaccuracy of computing the Shapley value.

**Inherent numerical instability in Linear TreeShap and TreeShap-K.** As detailed in Appendix B, Linear TreeShap involves inverting a Vandermonde matrix based on the Chebyshev nodes of the second kind. However, as proved by Tyrtyshnikov (1994, Theorem 4.1), the condition number of Vandermonde matrices based on real-valued nodes grows exponentially w.r.t. $D$. Therefore, Linear TreeShap potentially suffers from numerical instability when $D$ is large. Though TreeShap-K is free of Vandermonde matrices, it also faces such a numerical issue as it contains a step that essentially solves an ill-conditioned linear equation; see Appendix D for details. Figure 2 shows that the condition numbers of the involved matrices are already over $100,000$ when $D = 20$. Furthermore, it also concretely demonstrates how inaccurate Linear TreeShap would be. Refer to Appendix G for further experimental details and results.

**Well-conditioned Vandermonde matrices.** According to Pan (2016), a Vandermonde matrix of large size is ill-conditioned unless its nodes are more or less spaced equally on the unit circle. Precisely, let $\{\chi_k\}_{1 \le k \le n}$ be a list of complex-valued nodes where $\chi_k := e^{\frac{i2(k-1)\pi}{D}}$. Then, the condition number of the Vandermonde matrix $\mathbf{V} \in \mathbb{C}^{D \times D}$ defined by letting $V_{ij} := \chi_i^{j-1}$ is perfectly $1$, independent of $D$. Another nice properties are $\mathbf{V}^{-1} = \frac{1}{D}\overline{\mathbf{V}}$ and $\mathbf{V}^{\mathsf{T}} = \mathbf{V}$. Therefore, we will use this $\mathbf{V}$ for encoding and decoding polynomials in our TreeProb algorithm. We refer the reader to Appendix C for how TreeProb is implemented. Concurrently, Jiang et al. (2025) also observe this numerical issue and propose an identical solution when computing the Shapley value of $R^2$ for individual features. However, this solution is not fully numerically stable; it instead introduces numerical underflow when the number of nodes is large. As demonstrated in Appendix G, the numerical error of TreeProb still blows up as the number of nodes increases.

**Numerically accurate algorithms for computing weighted Banzhaf values and Beta Shapley values with integral parameters.** For the Shapley value, $\mu$ is the uniform distribution (Dubey et al., 1981, Theorem 1(a')), i.e., $\phi^{\text{Shap}}(f_{\mathbf{x}}) = \int_0^1 \nabla \overline{f}_{\mathbf{x}}(t \cdot \mathbf{1}_N)\mathrm{d}t$. Then, we have $\phi^{\text{Shap}}(f_{\mathbf{x}}) = \sum_{\ell=1}^{\lceil \frac{D}{2} \rceil} \kappa_\ell \cdot \nabla \overline{f}_{\mathbf{x}}(t_\ell \cdot \mathbf{1}_N)$ using the Gauss–Legendre quadrature rule. Since $\kappa_\ell > 0$ and $\sum_{\ell}^{\lceil \frac{D}{2} \rceil} \kappa_\ell = 1$, the Shapley value is just a weighted sum of finitely many gradients. Likewise, this argument extends to Beta Shapley values with integral parameters, i.e., $\text{Beta}(\alpha, \beta)$ with $\alpha, \beta \in \mathbb{N} \setminus \{0\}$. This observation immediately leads to our TreeGrad-Shap shown in Algorithms 6 and 7. Both algorithms run in $O(LD)$ time; the former consumes $O(L)$ space, while the latter vectorizes its for loop and therefore uses $O(D^2 + L)$ space. Note that TreeGrad-Shap inherits its numerical stability from TreeGrad, which computes weighted Banzhaf values.

**Corollary 1.** *TreeGrad-Shap (Algorithm 7) computes Beta Shapley value with integral parameters using $O(LD)$ time and $O(D^2 + L)$ space. Without vectorization (Algorithm 6), its space complexity is $O(L)$.*

# 6 EMPIRICAL RESULTS

**Metrics.** The goal of our experiments is to compare TreeGrad-Ranker with the Beta Shapley values using the insertion and deletion metrics (Petsiuk et al., 2018). We report the averaged insertion and

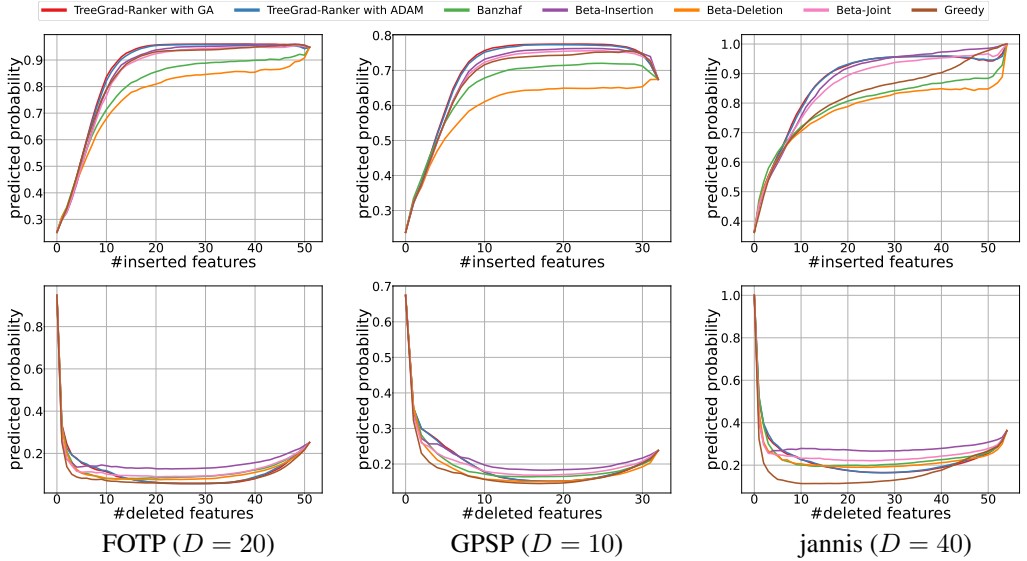

FOTP ($D = 20$)   GPSP ($D = 10$)   jannis ($D = 40$)

Figure 3: The trained models are **decision trees**. The first row shows the insertion curves, while the second presents the deletion curves. A higher insertion curve or a lower deletion curve indicates better performance. For our TreeGrad-Ranker, we set $T = 100$ and $\epsilon = 5$. Beta-Insertion selects the candidate Beta Shapley value that maximizes the $\text{Ins}$ metric for each $f_{\mathbf{x}}$, whereas Beta-Deletion minimizes the $\text{Del}$ metric. Beta-Joint selects the candidate that maximizes the joint metric $\text{Ins} - \text{Del}$. The output of each $f(\mathbf{x})$ is set to be the probability of **its predicted class**.

deletion curves and note that the values of $\text{Ins}$ and $\text{Del}$ defined in Eq. (4) correspond to the areas under the respective curves. Thus, in terms of increasing $f(\mathbf{x})$, a higher insertion curve indicates better ranking performance at the top, while a lower deletion curve reflects better ranking quality at the bottom.

**Datasets.** Our experiments are conducted using nine datasets from OpenML. Datasets of classification include: (1) first-order-theorem-proving (FOTP), (2) GesturePhaseSegmentationProcessed (GPSP), (3) jannis, (4) spambase, (5) philippine and (6) MinibooNE. Datasets of regression include: (7) Buzzinsocialmedia_Twitter (BT), (8) superconduct and (9) wave_energy. We randomly separate each dataset into a training dataset (80%) and a test dataset (20%). The training set is used to train decision trees, and each $\mathbf{x}$ is selected from the test set. For BT and wave_energy, we scale its predicted values for better presentation. A summary of the used datasets can be found in Table 1.

**Baselines.** Following Kwon & Zou (2022a;b), the selected baselines include Beta Shapley values $\text{Beta}(\alpha, \beta)$ with $(\alpha, \beta) \in \{(16, 1), (8, 1), (4, 1), (2, 1), (1, 1), (1, 2), (1, 4), (1, 8), (1, 16)\}$, and the Banzhaf value. Recall that $\text{Beta}(1, 1)$ corresponds to the Shapley value. To avoid noticeable numerical errors, they are computed using TreeGrad-Shap (Algorithm 6) and TreeGrad (Algorithm 1).[3] We also compare with a greedy algorithm. Starting from $S_0 := \emptyset$, the $t$-th top feature $\pi(t)$ is selected as $\arg\max_{i \in [N] \setminus S_{t-1}} f_{\mathbf{x}}(S_{t-1} \cup i) - f_{\mathbf{x}}([N] \setminus (S_{t-1} \cup i))$, and we update $S_t := S_{t-1} \cup \pi(t)$.

All decision trees are trained using the scikit-learn library (Pedregosa et al., 2011). To account for the variability of $f_{\mathbf{x}}$, we randomly select 200 instances of $\mathbf{x}$ to report the averaged results for each trained decision tree $f$. We fix the random seed to 2025 to ensure reproducibility. More details and results are deferred to the Appendix G.

For classification tasks, the output of each $f(\mathbf{x})$ is chosen to be the probability of either (i) its predicted class or (ii) a randomly sampled non-predicted class. All the results of decision trees are presented in Figures 3 and 4, with additional results provided in Appendix G. Our observations are as follows: (1) For Beta Shapley values, the one that achieves the best insertion metric often does not achieve the best deletion metric, and vice versa; (2) Our TreeGrad-Ranker significantly outperforms the selected baselines on the insertion metric while remaining competitive on the deletion metric.

---

[3]These two algorithms are combined into TreeStab in our GitHub repository.

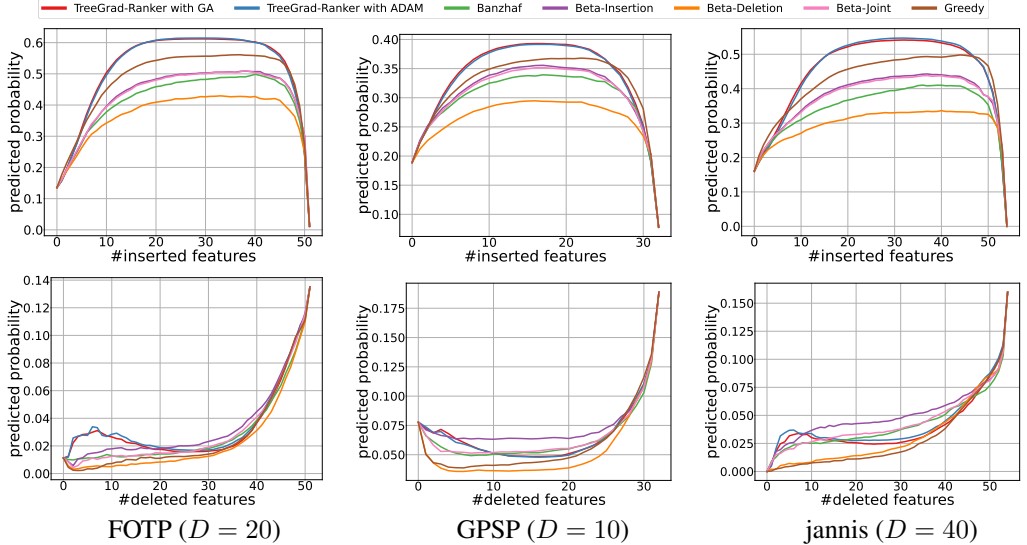

Figure 4: The trained models are **decision trees**. The first row shows the insertion curves, while the second presents the deletion curves. A higher insertion curve or a lower deletion curve indicates better performance. For our TreeGrad-Ranker, we set $T = 100$ and $\epsilon = 5$. Beta-Insertion selects the candidate Beta Shapley value that maximizes the Ins metric for each $f_{\mathbf{x}}$, whereas Beta-Deletion minimizes the Del metric. Beta-Joint selects the candidate that maximizes the joint metric Ins−Del. The output of each $f(\mathbf{x})$ is set to be the probability of **a randomly sampled non-predicted class**.

## 7 CONCLUSION

In this work, we revisit the evaluation of feature rankings through insertion and deletion metrics and demonstrate their connection to the problem (5). From this perspective, we establish theoretical limitations of probabilistic values, showing that they are no better than random in distinguishing patterns and that their reliance on linearization may render them ineffective. To address these shortcomings, we propose directly solving the problem (5) using gradient ascent. We developed Tree-Grad to compute these gradients for decision trees in $O(L)$ time. We show that solving problem (5) yields feature scores that satisfy all axioms uniquely characterizing probabilistic values, except linearity. Furthermore, our TreeGrad-Shap is the only numerically stable algorithm for computing Beta Shapley values with integral parameters. Our experiments show that TreeGrad-Ranker outperforms probabilistic values on both insertion and deletion metrics.

### ACKNOWLEDGMENT

This research is supported by the National Research Foundation (NRF), Prime Minister's Office, Singapore under its Campus for Research Excellence and Technological Enterprise (CREATE) programme. The Mens, Manus, and Machina (M3S) is an interdisciplinary research group (IRG) of the Singapore MIT Alliance for Research and Technology (SMART) centre. YY gratefully acknowledges NSERC and CIFAR for funding support.

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

# Table of Contents

## A  SEMI-VALUES

One popular way to rank features is by using the Shapley value, as it uniquely satisfies the axioms of linearity, null, symmetry, and efficiency (Shapley, 1953). Based on the Shapley value, the contribution of the $i$-th feature can be expressed as

$$\phi_i^{\text{Shap}}(f_{\mathbf{x}}) := \sum_{S \subseteq [N]\setminus i} \frac{s!(N-1-s)!}{N!} [f_{\mathbf{x}}(S \cup i) - f_{\mathbf{x}}(S)]$$

where $f_{\mathbf{x}}(S) := \mathbb{E}_{\mathbf{X}_{S^c}}[f(\mathbf{x}_S, \mathbf{X}_{S^c})]$ where $\mathbf{x}_S$ is the restriction of $\mathbf{x}$ on $S$ and $S^c := [N] \setminus S$. Following Lundberg et al. (2020, Algorithm 1), the computation of $f_{\mathbf{x}}(S)$ is described in Algorithm 8.

Recall that each probabilistic value can be expressed as

$$\phi_i^{\boldsymbol{\omega}}(f_{\mathbf{x}}) := \sum_{S \subseteq [N]\setminus i} \omega_{s+1} \cdot [f_{\mathbf{x}}(S \cup i) - f_{\mathbf{x}}(S)] \tag{9}$$

where $\boldsymbol{\omega} \in \mathbb{R}^N$ is non-negative with $\sum_{k=1}^N \binom{N-1}{k-1}\omega_k = 1$. Besides, $\phi^{\boldsymbol{\omega}}$ is a semi-value if and only if there exists a Borel probability measure $\mu$ over the closed interval $[0,1]$ such that for every $k \in [N]$

$$\omega_k = \int_0^1 t^{k-1}(1-t)^{N-k}\mathrm{d}\mu(t). \tag{10}$$

Therefore, $\phi^\mu(f_{\mathbf{x}})$ is used to denote semi-values. If $\mu$ is some Dirac measure $\delta_\nu$, the resulting semi-value is a weighted Banzhaf value parameterized by $\nu \in [0,1]$. If $\mu(A) \propto \int_A z^{\beta-1}(1-z)^{\alpha-1}\mathrm{d}z$, it leads to a Beta Shapley value parameterized by $(\alpha, \beta)$; in particular, the parameter $(1,1)$ leads to the Shapley value.

**Symmetric semi-values.**  A semi-value $\phi^\mu$ is said to be symmetric if $\omega_k = \omega_{N+1-k}$ for every $1 \le k \le N$. In particular, the Shapley and Banzhaf values are symmetric. Combining Eqs. (7), (9) and (10), we have

$$\phi^\mu(f_{\mathbf{x}}) = \int_0^1 \sum_{S \subseteq [N]\setminus i} t^k(1-t)^{N-1-k}[f_{\mathbf{x}}(S \cup i) - f_{\mathbf{x}}(S)]\,\mathrm{d}\mu(t)$$

$$= \int_0^1 \nabla \overline{f}_{\mathbf{x}}(t \cdot \mathbf{1}_N)\,\mathrm{d}\mu(t).$$

In particular, if $\phi^\mu$ is symmetric, we can derive

$$\phi^\mu(f_{\mathbf{x}}) = -\int_0^1 \nabla \overline{f}_{\mathbf{x}}((1-t) \cdot \mathbf{1}_N)\,\mathrm{d}\mu(t).$$

Therefore, for symmetric semi-values, we have

$$\phi^\mu(f_\mathbf{x}) = \int_0^1 \nabla \frac{1}{2}[\overline{f}_\mathbf{x}(t \cdot \mathbf{1}_N) - \overline{f}_\mathbf{x}((1-t) \cdot \mathbf{1}_N)]\,\mathrm{d}\mu(t).$$

In other words, symmetric semi-values can be derived from the gradient field of the objective in Eq. (6). This motivates averaging the gradients produced by our TreeGrad-Ranker (Algorithms 2 and 10) to generate feature scores.

## B  POLYNOMIAL MANIPULATIONS IN LINEAR TREESHAP

In manipulating polynomials, the most expensive step is the multiplication of $p(y)$ and $(1 + y)^{d-\deg(p)}$, the degree of each is at most $D$ in the context. In general, polynomial multiplication requires $O(D \log D)$ time to compute using fast Fourier transform (Cormen et al., 2022, Chapter 30), which would lead to time complexity $O(LD \log D)$ in total for computing $\phi^{\mathrm{Shap}}(f_\mathbf{x})$. Nevertheless, since the polynomial $(1 + y)^{d-\deg(p)}$ is simple, Yu et al. (2022) propose to encode all polynomials of degree at most $D - 1$ into vectors of dimension $D$ using a Vandermonde matrix $\mathbf{V} \in \mathbb{R}^{D \times D}$, i.e., each polynomial $p(y)$ is encoded by its evaluation $\mathbf{V}\mathbf{p}$ at $D$ different points, and then the multiplication can be computed in $O(D)$ time using the encoded vectors. Since the inverse of Vandermonde matrices can be computed once in $O(D^2)$ time (Eisinberg & Fedele, 2006), it does not increase the time complexity in total. Let $p(y)$ and $q(y)$ be two polynomials of degree at most $D-1$, by Yu et al. (2022, Lemma 2.5), the formula used to evaluate $\psi$ is $\langle p(y), q(y) \rangle = \langle \mathbf{p}, \mathbf{q} \rangle = \langle \mathbf{V}\mathbf{p}, (\mathbf{V}^{-1})^\mathsf{T}\mathbf{q} \rangle.$.

**Numerical Instability.**  Empirically, we observe that numerically error grows as the number of nodes increases. To mitigate this numerical instability, the implementation of Linear TreeShap does not follow the $O(LD)$-time procedure.[4]  We illustrate this difference using Algorithm 3. Upon reaching a leaf (line 46), Linear TreeShap computes $p(y) \leftarrow \frac{\varrho_v c_v}{c_r} \cdot p(y)$. In line 48, it performs

$$p_a(y) \leftarrow \mathrm{traverse}(a_v, p(y)), \quad p_b(y) \leftarrow \mathrm{traverse}(b_v, p(y)), \quad d_{\max} \leftarrow \max(\deg(p_a(y)), \deg(p_b(y)))$$
$$p(y) \leftarrow p_a(y) \cdot (1+y)^{d_{\max}-\deg(p_a(y))} + p_b(y) \cdot (1+y)^{d_{\max}-\deg(p_b(y))}$$

As a result, when evaluating $\psi$ in line 52, the degree of $G[v]$ is not fixed, and may vary from 1 to $D$. Consequently, given $D$ nodes, the algorithm has to compute $\mathbf{V}_d^{-1}$ for every $d \in [D]$, where $\mathbf{V}_d \in \mathbb{R}^{d \times d}$ denotes the Vandermonde matrix of the first $d$ nodes. This effectively reduces the number of nodes involved in each computation and thus partially alleviates the inherent numerical instability. However, this strategy increases the time complexity from $O(LD)$ to $O(D^3 + LD)$.

## C  IMPLEMENTATION OF TREEPROB

Similar to Linear TreeShap, our TreeProb is composed of two procedures: (i) a traversing-down procedure that for every leaf node $v \in L_r$ accumulates all the polynomial components from the root to $v$ one by one and (ii) a traversing-up procedure that decodes the accumulated polynomials and distributes the decoded results to all participating features. Starting from the root $r$, write $P_v = (e_1, e_2, \ldots, e_{|P_v|})$ in the order in which the edges are encountered as we traverse down the path $P_v$ to the leaf node $v$. Then, TreeProb accumulates the polynomial components as follows:

$$p_0(y) \leftarrow 1 \quad \text{and} \quad p_\ell(y) \leftarrow \begin{cases} \frac{p_{\ell-1}(y)}{1 + \gamma_{e_\ell^\uparrow} \cdot y} \cdot (1 + \gamma_{e_\ell} \cdot y), & e_\ell^\uparrow \text{ exists,} \\ p_{\ell-1}(y) \cdot (1 + \gamma_{e_\ell} \cdot y), & \text{otherwise.} \end{cases}$$

Once it reaches the leaf node $v$, there is $\frac{\varrho_v c_v}{c_r} \cdot p_{|P_v|}(y) = G_v(y)$. While traversing up, the formula we use is

$$\phi_i^{\boldsymbol{\omega}}(f_\mathbf{x}) = \sum_{\substack{e \in E \\ l_e = i}} (\gamma_e - 1)\langle \frac{\sum_{v \in L_{h_e^\dagger}} G_v(y) \cdot (1+y)^{D-\deg(G_v)}}{1 + \gamma_e \cdot y}, q_{D-1}^{\boldsymbol{\omega}}(y) \rangle.$$

---

[4] https://github.com/yupbank/linear_tree_shap/blob/main/linear_tree_shap/fast_linear_tree_shap.py

---

**Algorithm 3** TreeProb

---

**Input:** Decision tree $\mathcal{T} = (V, E)$ with root $r$ and depth $D$, instance $\mathbf{x} \in \mathbb{R}^N$, polynomial $q(y)$ of degree $\min(D, N) - 1$ with non-negative coefficients that satisfies $\sum_{k=0}^{\min(D,N)-1} \binom{\min(D,N)-1}{k} q_{k+1} = 1$

**Output:** Probabilistic value $\phi$

28  **Function** `traverse`$(v, p(y) = 1)$**:**

29  $\quad e \leftarrow (m_v \rightarrow v)$

30  $\quad$ **if** $e^{\uparrow}$ *exists* **then**

31  $\quad\quad\quad p(y) \leftarrow \dfrac{p(y)}{1 + \gamma_{e^{\uparrow}} \cdot y} \cdot (1 + \gamma_e \cdot y)$

32  $\quad$ **else**

33  $\quad\quad\quad p(y) \leftarrow p(y) \cdot (1 + \gamma_e \cdot y)$

34  $\quad$ **if** $v$ *is a leaf* **then**

35  $\quad\quad\quad p(y) \leftarrow \frac{\varrho_v c_v}{c_r} \cdot p(y) \cdot (1 + y)^{\min(D,N) - \deg(p)}$

36  $\quad$ **else**

37  $\quad\quad\quad p(y) \leftarrow$ `traverse`$(a_v, p(y)) +$ `traverse`$(b_v, p(y))$

38  $\quad$ **if** $e^{\uparrow}$ *exists* **then**

39  $\quad\quad\quad G[h_{e^{\uparrow}}] \leftarrow G[h_{e^{\uparrow}}] - p(y)$

40  $\quad$ $G[v] \leftarrow G[v] + p(y)$

41  $\quad$ $\phi_{l_e} \leftarrow \phi_{l_e} + (\gamma_e - 1) \cdot \left\langle \frac{G[v]}{1 + \gamma_e \cdot y}, q(y) \right\rangle$

42  $\quad$ **return** $p(y)$

43  Initialize $G[v] = 0$ for every $v \in V$

44  Initialize $\phi = \mathbf{0}_N$

45  `traverse`$(a_r)$

46  `traverse`$(b_r)$

47  **return** $\phi$

---

where

$$q_{D-1}^{\boldsymbol{\omega}}(y) = \sum_{k=0}^{D-1} \left( \sum_{j=0}^{N-D} \binom{N-D}{j} \omega_{k+j+1} \right) y^k, \tag{11}$$

which is derived by recursively applying the equality in Lemma 3. At first glance, this formula may appear complicated; however, it can be efficiently implemented using lines 47–51 of Algorithm 3. For semi-values $\phi^{\mu}(f_{\mathbf{x}})$, we write $q_{D-1}^{\mu}(y)$ instead, which can be computed using Eq. (8).

**Remark 1.** *Note that Eq. (11) is well-defined when $N \geq D$. However, if $N < D$, we can simply set $D \leftarrow N$. By contrast, Eq. (8) is always well-defined. Therefore, we recommend applying $D \leftarrow \min(D, N)$ before running Algorithm 3.*

**Remark 2.** *For probabilistic values, it is more convenient to specify $q_{D-1}^{\boldsymbol{\omega}}(y)$ instead of deriving it from $\boldsymbol{\omega}$. In particular, there must be $\sum_{k=0}^{D-1} \binom{D-1}{k} q_{k+1} = 1$ where $q_k$ is the $k$-th coefficient of $q_{D-1}^{\boldsymbol{\omega}}(y)$.*

The pseudo-code of our TreeProb is presented in Algorithm 3. One may think that the space complexity $O(LD)$. We remark that there are only $O(D)$ polynomials that are needed on the spot and thus the complexity is $O(D^2 + L)$. Therefore, for computing probabilistic values, the time and space complexities of TreeProb are $O(LD)$ and $O(D^2 + L)$, respectively.

## D    HOW TO EVOLVE LINEAR TREESHAP INTO TREESHAP-K

**How to avoid the use of Vandermonde matrices?** As discussed in Appendix B, the role of Vandermonde matrices is to save the computation time of the scaling $G_v(y) \cdot (1 + y)^{d - \deg(G_v)}$ performed at every non-leaf node from $O(D \log D)$ to $O(D)$. Our observation is that such a scaling can be

performed just once for all $G_v(y)$. Starting from the root, write $P_v = (e_1, e_2, \ldots, e_{|P_v|})$ in the order encountered when traversing down the path $P_v$ to the leaf node $v$. A modified procedure for collecting the polynomial components is

$$p_0(y) \leftarrow (1+y)^D \ \text{ and } \ p_\ell(y) \leftarrow \begin{cases} \dfrac{p_{\ell-1}(y)}{1 + \gamma_{e_\ell^\uparrow} \cdot y} \cdot (1 + \gamma_{e_\ell} \cdot y), & e_\ell^\uparrow \text{ exists,} \\ \dfrac{p_{\ell-1}(y)}{1+y} \cdot (1 + \gamma_{e_\ell} \cdot y), & \text{otherwise.} \end{cases}.$$

A similar technique is employed in Marichal & Mathonet (2011, Appendix E), and thus their proposed algorithm is free of Vandermonde matrices. It is worth pointing out that the assignment of $p_0(y)$ is performed once and costs $O(D^2)$ time in a recursive way.

By traversing down in this way, we can store each polynomial $p(y) = \sum_{k=0}^d p_{k+1} y^k$ as its coefficient vector $\mathbf{p} \in \mathbb{R}^{D+1}$. For updating $q(y) \leftarrow p(y) \cdot (1 + \gamma \cdot y)$, we simply run $q_k = p_k + \gamma \cdot p_{k-1}$ with $p_{-1} = p_{\deg(p)+2} = 0$. Note that this process costs $O(D)$ time and can be reversed in $O(D)$ time. However, this reverse process is essentially solving a linear equation, and we observe empirically that the corresponding condition number grows exponentially w.r.t. $D$ if $\gamma \geq 1$.

**How TreeShap-K computes the Shapley value.** What TreeShap-K stores, instead, is a vector $\boldsymbol{\vartheta}^v \in \mathbb{R}^{D+1}$ where $\vartheta_i^v$ represents the $i$-th coefficient of $G_v(y) \cdot (1+y)^{D-|F_v|}$. If $D = |F_v|$, $\vartheta_i^v = \frac{\varrho_v c_v}{c_r} \sum_{S \subseteq F_v}^{|S|=k} \prod_{i \in S} \gamma_{i,v}$. Then, while traversing down, each $\boldsymbol{\vartheta}^v$ can be computed by

$$\boldsymbol{\theta}_0 \leftarrow \mathbf{1}_{D+1} \ \text{ and } \ \boldsymbol{\theta}_\ell \leftarrow \begin{cases} (\boldsymbol{\theta}_{\ell-1} \ominus \gamma_{e_\ell^\uparrow}) \oplus \gamma_{e_\ell}, & e_\ell^\uparrow \text{ exists,} \\ (\boldsymbol{\theta}_{\ell-1} \ominus 1) \oplus \gamma_{e_\ell}, & \text{otherwise.} \end{cases}$$

Here, suppose $\boldsymbol{\xi} \in \mathbb{R}^{D+1}$, for $\boldsymbol{\varphi} \leftarrow \boldsymbol{\xi} \oplus \gamma$, it is

$$\varphi_k \leftarrow \frac{D+1-k}{D+1}\xi_k + \gamma \cdot \frac{k-1}{D+1}\xi_{k-1}, \tag{12}$$

whereas $\boldsymbol{\varphi} \ominus \gamma$ denotes the reverse process. The convention is $\xi_{-1} = \xi_{D+1} = 0$. Then, $\frac{\varrho_v c_v}{c_r} \cdot \boldsymbol{\theta}_{|P_v|} = \boldsymbol{\vartheta}^v$. What TreeShap-K computes in the traversing-up procedure is

$$\phi_i^{\text{Shap}}(f_\mathbf{x}) = \sum_{\substack{e \in E \\ l_e = i}} (\gamma_e - 1) \cdot \text{sum}\Big(\sum_{v \in L_{h_e}^\dagger} \boldsymbol{\vartheta}^v \ominus \gamma_e\Big)$$

where $\text{sum}$ simply adds up all the entries. All in all, TreeShap-K proposed by Karczmarz et al. (2022, Appendix E) is summarized in Algorithm 4. We point out that the $\ominus$ operation essentially involves solving a linear equation, whose condition number, as shown in Figure 2, grows exponentially w.r.t. $D$ (with $\gamma$ set to 1). Indeed, the substantial inaccuracy of TreeShap-K is already confirmed by Karczmarz et al. (2022, Figure 1).

# E ONE MORE $O(LD)$-TIME AND $O(D^2 + L)$-SPACE ALGORITHM FOR COMPUTING THE SHAPLEY VALUE

For completeness, we also summarize the technical details of one more $O(LD)$-time and $O(D^2 + L)$-space algorithm for computing the Shapley value, which was developed by Yu et al. (2022) as their first version.[5] Therefore, we refer to this algorithm as Linear TreeShap V1. It is totally different from Linear TreeShap, and is less numerically stable.[6] To our knowledge, its technical details are missing, probably due to its numerical instability. Nevertheless, it is with a moment's thought to

---

[5]https://github.com/yupbank/linear_tree_shap/blob/main/linear_tree_shap/linear_tree_shap_numba.py. The code therein cannot run, but it can be fixed by replacing the first three `tree.max_depth` with `tree.max_depth + 1` in lines 145-146.

[6]https://github.com/yupbank/linear_tree_shap/blob/main/tests/correctness.csv.

---

**Algorithm 4** TreeShap-K

---

**Input:** Decision tree $\mathcal{T} = (V, E)$ with root $r$ and depth $D$, instance $\mathbf{x} \in \mathbb{R}^N$
**Output:** Shapley value $\phi$

48 **Function** `traverse` $(v, \boldsymbol{\theta} = \frac{1}{M+1}\mathbf{1}_{M+1})$**:**

49 $\quad$ $e \leftarrow (m_v \rightarrow v)$

50 $\quad$ **if** $e^{\uparrow}$ *exists* **then**

51 $\quad$ $\quad$ $\boldsymbol{\theta} \leftarrow (\boldsymbol{\theta} \ominus \gamma_{e^{\uparrow}}) \oplus \gamma_e$

52 $\quad$ **else**

53 $\quad$ $\quad$ $\boldsymbol{\theta} \leftarrow (\boldsymbol{\theta} \ominus 1) \oplus \gamma_e$

54 $\quad$ **if** $v$ *is a leaf* **then**

55 $\quad$ $\quad$ $\boldsymbol{\theta} \leftarrow \frac{\varrho_v c_v}{c_r} \cdot \boldsymbol{\theta}$

56 $\quad$ **else**

57 $\quad$ $\quad$ $\boldsymbol{\theta} \leftarrow$ `traverse` $(a_v, \boldsymbol{\theta}) +$ `traverse` $(b_v, \boldsymbol{\theta})$

58 $\quad$ **if** $e^{\uparrow}$ *exists* **then**

59 $\quad$ $\quad$ $\vartheta[h_{e^{\uparrow}}] \leftarrow \vartheta[h_{e^{\uparrow}}] - \boldsymbol{\theta}$

60 $\quad$ $\vartheta[v] \leftarrow \vartheta[v] + \boldsymbol{\theta}$

61 $\quad$ $\phi_{l_e} \leftarrow \phi_{l_e} + (\gamma_e - 1) \cdot \text{sum}(\vartheta[v] \ominus \gamma_e)$

62 $\quad$ **return** $\boldsymbol{\theta}$

63 $M \leftarrow \min(D, N)$

64 Initialize $\vartheta[v] = \mathbf{0}_{M+1}$ for every $v \in V$

65 Initialize $\phi = \mathbf{0}_N$

66 `traverse` $(a_r)$

67 `traverse` $(b_r)$

68 **return** $\phi$

---

decipher Linear TreeShap V1 from the code by observing that it uses $\gamma_e - 1$ rather than $\gamma_e$ when traversing down the tree. It is based on the fact that

$$\phi_i^{\text{Shap}}(f_{\mathbf{x}}) = \int_0^1 E_i(t) \, \mathrm{d}t$$

$$\text{where} \quad E_i(t) := \sum_{S \subseteq [N] \setminus i} t^s (1-t)^{N-1-s} [f_{\mathbf{x}}(S \cup i) - f_{\mathbf{x}}(S)].$$

Since $E_i(t)$ is a polynomial of degree at most $N - 1$, we can express $E_i(t)$ as

$$E_i(t) = \sum_{d=0}^{N-1} \alpha_{i,d} \cdot t^d.$$

Then,

$$\phi_i^{\text{Shap}}(f_{\mathbf{x}}) = \sum_{d=0}^{N-1} \frac{\alpha_{i,d}}{d+1}.$$

**Lemma 4.** *Replacing $f_{\mathbf{x}}$ with $R_{\mathbf{x}}^v$, where $R_{\mathbf{x}}^v(S) = \varrho_v \cdot \frac{c_v}{c_r} \prod_{j \in S} \gamma_{j,v}$ for every $S \subseteq [N]$, we have*

$$\alpha_{i,d} = \varrho_v(\gamma_{i,v} - 1) \cdot \frac{c_v}{c_r} \sum_{\substack{S \subseteq [N] \setminus i \\ s = d}} \prod_{j \in S} (\gamma_{j,v} - 1).$$

*Proof.* Observe that

$$E_i(t) = \varrho_v(\gamma_{i,v} - 1) \cdot \frac{c_v}{c_r} \sum_{S \subseteq [N] \setminus i} t^s (1-t)^{N-1-s} \prod_{j \in S} \gamma_{j,v}$$

$$= \varrho_v(\gamma_{i,v} - 1) \cdot \frac{c_v}{c_r} \prod_{j \in [N] \setminus i} (1 + t \cdot (\gamma_{j,v} - 1)),$$

from which the conclusion follows. $\qquad\square$

---

**Algorithm 5** Linear TreeShap V1

**Input:** Decision tree $\mathcal{T} = (V, E)$ with root $r$ and depth $D$, instance $\mathbf{x} \in \mathbb{R}^N$
**Output:** Shapley value $\phi$

69 **Function** `traverse`$(v, \mathbf{c} = \mathbf{e}_1)$**:**
70 $\quad e \leftarrow (m_v \to v)$
71 $\quad$ **if** $e^\uparrow$ *exists* **then**
72 $\quad\quad \mathbf{c} \leftarrow (\mathbf{c} \boxminus (\gamma_{e^\uparrow} - 1)) \boxplus (\gamma_e - 1)$
73 $\quad$ **else**
74 $\quad\quad \mathbf{c} \leftarrow \mathbf{c} \boxplus (\gamma_e - 1)$
75 $\quad$ **if** $v$ *is a leaf* **then**
76 $\quad\quad \mathbf{c} \leftarrow \frac{\varrho_v c_v}{c_r} \cdot \mathbf{c}$
77 $\quad$ **else**
78 $\quad\quad \mathbf{c} \leftarrow$ `traverse`$(a_v, \mathbf{c})$ + `traverse`$(b_v, \mathbf{c})$
79 $\quad$ **if** $e^\uparrow$ *exists* **then**
80 $\quad\quad \mathcal{C}[h_{e^\uparrow}] \leftarrow \mathcal{C}[h_{e^\uparrow}] - \mathbf{c}$
81 $\quad \mathcal{C}[v] \leftarrow \mathcal{C}[v] + \mathbf{c}$
82 $\quad \phi_{l_e} \leftarrow \phi_{l_e} + (\gamma_e - 1) \cdot \sum_{k=1}^{M+1} \frac{(\mathcal{C}[v] \boxminus (\gamma_e - 1))_k}{k}$
83 $\quad$ **return** $\mathbf{c}$

84 $M \leftarrow \min(D, N)$
85 Initialize $\mathcal{C}[v] = \mathbf{0}_{M+1}$ for every $v \in V$
86 Initialize $\phi = \mathbf{0}_N$
87 `traverse`$(a_r)$
88 `traverse`$(b_r)$
89 **return** $\phi$

---

Accordingly, after Linear TreeShap V1 traverses down the path $P_v = (e_1, e_2, \ldots, e_{|P_v|})$, it stores a vector $\boldsymbol{\iota}^v \in \mathbb{R}^{D+1}$ where $\iota_k^v = \sum_{\substack{S \subseteq F_v \\ s = k-1}} \prod_{j \in S}(\gamma_{j,v} - 1)$ if $1 \leq k \leq |F_v| + 1$ and 0 otherwise.

One nice property is that, after introducing a null player $k$ (with $\gamma_{k,v} = 1$), $\boldsymbol{\iota}^v \in \mathbb{R}^{D+1}$ remains unchanged. All in all, while traversing down, each $\boldsymbol{\iota}^v$ can be computed by

$$\mathbf{c}_0 \leftarrow \mathbf{e}_1 \quad \text{and and} \quad \mathbf{c}_\ell \leftarrow \begin{cases} (\mathbf{c}_{\ell-1} \boxminus (\gamma_{e_\ell^\uparrow} - 1)) \boxplus (\gamma_{e_\ell} - 1), & e_\ell^\uparrow \text{ exists,} \\ \mathbf{c}_{\ell-1} \boxplus (\gamma_{e_\ell} - 1), & \text{otherwise.} \end{cases}$$

where $\mathbf{e}_1 \in \mathbb{R}^{D+1}$ denotes the vector whose first entry is equal to 1 and all other entries are equal to 0. Here, suppose $\mathbf{d} \in \mathbb{R}^{D+1}$, for $\mathbf{c} \leftarrow \mathbf{d} \boxplus (\gamma - 1)$, it is

$$c_k = (\gamma - 1) \cdot d_{k-1} + d_k$$

whereas $\boldsymbol{\varphi} \boxminus \gamma$ denotes the reverse process. The convention is $d_{-1} = 0$. Then, $\boldsymbol{\iota}^v = \varrho_v \cdot \frac{c_v}{c_r} \mathbf{c}_{|P_v|}$. Algorithm 5 demonstrates how Linear TreeShap V1 runs. In our inspection, we observe that the condition number of $\boxminus$ does not even exceed 100 when $D = 100$. However, its numerical error still blows up, which we suspect is due to the repeated use of $\boxminus$.

## F MISSING PROOFS

**Proposition 2.** *Let $\pi$ be a ranking induced by a probabilistic value $\phi^{\boldsymbol{\omega}}(f_\mathbf{x})$ such that $\phi_{\pi(1)}^{\boldsymbol{\omega}}(f_\mathbf{x}) \geq \phi_{\pi(2)}^{\boldsymbol{\omega}}(f_\mathbf{x}) \geq \cdots \geq \phi_{\pi(N)}^{\boldsymbol{\omega}}(f_\mathbf{x})$. For every $1 \leq k \leq N$, $S_k^+ := \{\pi(1), \pi(2), \ldots, \pi(k)\}$ is an optimal solution to the problem*

$$\underset{S \subseteq [N], |S| = k}{\text{maximize}} \frac{1}{2} \left( g^*(S) - g^*([N] \setminus S) \right)$$

*where $g^*$ is the best linear surrogate for $f_\mathbf{x}$, i.e., $g^* = \arg\min_{g \in \mathcal{L}} \sum_{S \subseteq [N]} \eta_{s+1}(f_\mathbf{x}(S) - g(S))^2$ with $\mathcal{L} := \{g : g(S) = t_0 + \sum_{i \in S} t_i\}$. The weights satisfy (i) $\eta_1, \eta_{N+1} \geq 0$ and (ii) $\eta_s = \omega_s + \omega_{s-1}$ for $2 \leq s \leq N$.*

*Proof.* As provided by Li & Yu (2024b, Theorem 2), there exists a scalar $C \in \mathbb{R}$ such that for every $i \in [N]$

$$g^*(\{i\}) - g^*(\emptyset) = \phi_i^{\boldsymbol{\omega}}(f_{\mathbf{x}}) + C \cdot \mathbf{1}_N$$

where $\mathbf{1}_N \in \mathbb{R}^N$ is the all-one vector. Then, it is clear that $S_k^+$ is optimal to the problem

$$\arg\max_{S \subseteq [N], |S|=k} \frac{1}{2}(g^*(S) - g^*([N] \setminus S)),$$

by which the conclusion follows. $\square$

**Proposition 1.** *Suppose $\phi(f_{\mathbf{x}}) \in \mathbb{R}^N$ is linear in $f_{\mathbf{x}}$ with $N > 3$, which includes all probabilistic values. Then, there exist distinct sets $S_1$ and $S_2$ such that $S_1 \neq [N] \setminus S_2$ and*

$$\mathrm{Rel}(h; S_1, S_2) := \inf_{U: D_U(S_1) \leq D_U(S_2)} [1 - h(\phi(U))] + \inf_{U: D_U(S_1) > D_U(S_2)} h(\phi(U)) \leq 1.$$

*Proof.* This result is immediate if we can find two distinct subsets $S_1$ and $S_2$ satisfying $S_1 \neq [N] \setminus S_2$, and utility functions $U_0$ and $U_1$, such that $\phi^{\boldsymbol{\omega}}(U_0) = \phi^{\boldsymbol{\omega}}(U_1)$, $D_{U_0}(S_1) \leq D_{U_0}(S_2)$, and $D_{U_1}(S_1) > D_{U_1}(S_2)$.

Observe that each $U$ can be represented as a vector $\mathbf{x}_U \in \mathbb{R}^{2^N}$ whose entries correspond to all subsets of $[N]$. Since $\phi$ is a linear operator, its null space $\mathcal{N}$ has dimension at least $2^N - N$. It is sufficient to find a vector $\mathbf{x} \in \mathcal{N}$, for which there exists two distinct subsets $S$ and $T$ such that $S \neq [N] \setminus T$ and $|\mathrm{sign}(x_S - x_{[N] \setminus S}) - \mathrm{sign}(x_T - x_{[N] \setminus T})| = 2$.

Let $\mathcal{P} = \{\{S, [N] \setminus S\}: S \subseteq [N]\}$ and then $|\mathcal{P}| = 2^{N-1}$. For each $\{S, [N] \setminus S\} \in \mathcal{P}$, let $\mathrm{Choice}(\{S, [N] \setminus S\})$ randomly choose one of the set it contains. Then, let $\mathcal{I} = \{\mathrm{Choice}(\mathcal{S}): \mathcal{S} \in \mathcal{P}\}$ and we have $|\mathcal{I}| = 2^{N-1}$. Define a linear operator $L$ that maps each $\mathbf{x}$ to $\mathbf{y} \in \mathbb{R}^{2^{N-1}}$ by setting $y_S = x_S - x_{[N] \setminus S}$ for every $S \in \mathcal{I}$. Notice that the dimension of the null space of $T$ is exactly $2^{N-1}$. Denote the null space of $L$ by $\mathrm{null}\, L$ and write $\dim$ for dimension. By the rank–nullity theorem, there is

$$\dim \mathcal{N} = \dim \mathrm{null}\, L|_{\mathcal{N}} + \dim L(\mathcal{N}).$$

Therefore,

$$\dim L(\mathcal{N}) = \dim \mathcal{N} - \dim \mathrm{null}\, L|_{\mathcal{N}} \geq 2^N - N - 2^{N-1} = 2^{N-1} - N.$$

Given that $N > 3$, we have $\dim L(\mathcal{N}) > 2$. It indicates that we can find an $\mathbf{x} \in \mathcal{N}$ such that at least two entries of $\mathbf{y} := L(\mathbf{x})$, say $S$ and $T$, are nonzero. If $\mathrm{sign}(y_S) = \mathrm{sign}(y_T)$, set $T \leftarrow [N] \setminus T$. $\square$

**Theorem 1.** *The vector of feature scores $\boldsymbol{\zeta}$ returned by TreeGrad-Ranker, either using Algorithm 2 or 10, can be expressed as $\zeta_i = \sum_{S \subseteq [N] \setminus i} \omega_i(S; \mathbf{x})(f_{\mathbf{x}}(S \cup i) - f_{\mathbf{x}}(S))$ where $\omega_i(S; \mathbf{x}) > 0$ and $\sum_{S \subseteq [N] \setminus i} \omega_i(S; \mathbf{x}) = 1$. Moreover, it verifies the axioms of dummy, equal treatment and monotonicity.*

Let $U : 2^{[N]} \to \mathbb{R}$ be a utility function and denote by $\mathcal{G}$ the set that contains all such utility functions.

The axioms of interest are listed in the below.

1. **Linearity**: $\phi(c \cdot U + U') = c \cdot \phi(U) + \phi(U')$ for every $U, U' \in \mathcal{G}$ and $c \in \mathbb{R}$.

2. **Equal Treatment**: For some $i, j \in [N]$, if $U(S \cup i) = U(S \cup j)$ for every $S \subseteq [N] \setminus \{i, j\}$, then $\phi_i(U) = \phi_j(U)$.

3. **Symmetry**: $\phi(U \circ \pi) = \phi(U) \circ \pi$ for every permutation $\pi$ of $[N]$.

4. **Dummy**: For some $i \in [N]$, if there exists some $C \in \mathbb{R}$ such that $U(S \cup i) = U(S) + C$ for every $S \subseteq [N] \setminus i$. Then, $\phi_i(U) = C$.

5. **Monotonicity**: For some $i \in [N]$, (i) if $U(S \cup i) \geq U(S)$ for every $S \subseteq [N] \setminus i$, then $\phi_i(U) \geq 0$; (ii) if $U(S \cup i) \leq U(S)$ for every $S \subseteq [N] \setminus i$, then $\phi_i(U) \leq 0$.

6. **Efficiency**: $\sum_{i \in [N]} \phi_i(U) = U([N]) - U(\emptyset)$ for every $U \in \mathcal{G}$.

Setting $C = 0$ in the dummy axiom yields the null axiom. As aforementioned, the axioms of linearity, dummy, monotonicity and symmetry uniquely characterize the family of probabilistic values. By Li & Yu (2023, Appendix A), the symmetry axiom is equivalent to the axiom of equal treatment in the context.

*Proof.* Combing Eqs. (7) and (14), each $\nabla \overline{f}_{\mathbf{x}}(\mathbf{z})$ can be expressed as

$$\nabla_i \overline{f}_{\mathbf{x}}(\mathbf{z}) = \sum_{S \subseteq [N] \setminus i} \omega_i(S)(f_{\mathbf{x}}(S \cup i) - f_{\mathbf{x}}(S))$$

where $\omega_i(S) \geq 0$ and $\sum_{S \subseteq [N] \setminus i} \omega_i(S) = 1$ for every $i \in [N]$. Since $\boldsymbol{\zeta}$ returned by Algorithm 2 is an average of gradients, it is clear that for every $i \in [N]$,

$$\zeta_i = \sum_{S \subseteq [N] \setminus i} \omega_i(S; \mathbf{x})(f_{\mathbf{x}}(S \cup i) - f_{\mathbf{x}}(S))$$

where $\omega_i(S; \mathbf{x}) \geq 0$ and $\sum_{S \subseteq [N] \setminus i} \omega_i(S; \mathbf{x}) = 1$. The dependence of $\mathbf{x}$ comes from that different instances of $\mathbf{x}$ would induce different trajectories along which the gradients are accumulated and averaged. Suppose there exist a feature $i \in [N]$ and a constant $C \in \mathbb{R}$ such that $f_{\mathbf{x}}(S \cup i) = f_{\mathbf{x}}(S) + C$ for every $S \subseteq [N] \setminus i$. In this case,

$$\nabla_i \overline{f}_{\mathbf{x}}(\mathbf{z}) = C \cdot \sum_{S \subseteq [N] \setminus i} \omega_i(S) = C \ \text{ for every } \mathbf{z} \in [0,1]^N.$$

On average, it is clear that $\zeta_i = C$. Therefore, $\boldsymbol{\zeta}$ satisfies the dummy axiom. Similarly, one can verify that $\boldsymbol{\zeta}$ satisfies the monotonicity axiom.

Suppose there exist two features $i$ and $j$ such that $f_{\mathbf{x}}(S \cup i) = f_{\mathbf{x}}(S \cup j)$ for every $S \subseteq [N] \setminus \{i, j\}$. Observe that $\boldsymbol{\zeta}$ are averaged over $T$ gradients. At the $t$-th step, we denote the produced gradient by $g(\mathbf{z}^{(t-1)})$. Note that we set $\mathbf{z}^{(0)} = 0.5 \cdot \mathbf{1}_N$, which corresponds to the Banzhaf value. Therefore, we already have $g_i(\mathbf{z}^{(0)}) = g_j(\mathbf{z}^{(0)})$. The next step is to prove by induction that (i) $g_i(\mathbf{z}^{(t-1)}) = g_j(\mathbf{z}^{(t-1)})$ and $z_i^{(t-1)} = z_j^{(t-1)}$ (ii) for every $t$. Suppose it is true at the $t$-th step, for the next step, we have

$$z_i^{(t)} = z_i^{(t-1)} + g_i(\mathbf{z}^{(t-1)}) = z_j^{(t-1)} + g_j(\mathbf{z}^{(t-1)}) = z_j^{(t)}.$$

Let $w(S; \mathbf{z}) := \prod_{i \in S} z_i$, $m_i(S; \mathbf{z}) := \prod_{j \notin S \cup i}(1 - z_j)$ and $\Delta_i f_{\mathbf{x}}(S) := f_{\mathbf{x}}(S \cup i) - f_{\mathbf{x}}(S)$. For simplicity, we only consider the gradients of $\overline{f}_{\mathbf{x}}(\mathbf{z})$. The other part $\overline{f}_{\mathbf{x}}(\mathbf{1}_N - \mathbf{z})$ can be trivially included. Then,

$$g_i(\mathbf{z}^{(t)})$$
$$= \sum_{S \subseteq [N] \setminus i} w(S; \mathbf{z}^{(t)}) m_i(S; \mathbf{z}^{(t)}) \Delta_i f_{\mathbf{x}}(S)$$
$$= \sum_{S \subseteq [N] \setminus \{i,j\}} w(S; \mathbf{z}^{(t)}) m_i(S; \mathbf{z}^{(t)}) \Delta_i f_{\mathbf{x}}(S) + \sum_{S \subseteq [N] \setminus \{i,j\}} w(S \cup j; \mathbf{z}^{(t)}) m_i(S \cup j; \mathbf{z}^{(t)}) \Delta_i f_{\mathbf{x}}(S \cup j)$$
$$= \sum_{S \subseteq [N] \setminus \{i,j\}} w(S; \mathbf{z}^{(t)}) m_j(S; \mathbf{z}^{(t)}) \Delta_j f_{\mathbf{x}}(S) + \sum_{S \subseteq [N] \setminus \{i,j\}} w(S \cup i; \mathbf{z}^{(t)}) m_j(S \cup i; \mathbf{z}^{(t)}) \Delta_j f_{\mathbf{x}}(S \cup i)$$
$$= \sum_{S \subseteq [N] \setminus j} w(S; \mathbf{z}^{(t)}) m_j(S; \mathbf{z}^{(t)}) \Delta_j f_{\mathbf{x}}(S)$$
$$= g_j(\mathbf{z}^{(t)}).$$

Since $g_i(\mathbf{z}^{(t-1)}) = g_j(\mathbf{z}^{(t-1)})$ for every $t$, we have $\zeta_i = \zeta_j$. Consequently, $\boldsymbol{\zeta}$ satisfies the axiom of equal treatment. For Algorithm 10, it can be proved similarly. $\square$

**Lemma 3** (Generalization of Eq. (3)). *Suppose $p(y)$ is a polynomial of degree $N - 2$, we have* $\langle p(y) \cdot (1 + y), \sum_{k=0}^{N-1} \omega_{k+1} y^k \rangle = \langle p(y), \sum_{k=0}^{N-2} (\omega_{k+1} + \omega_{k+2}) y^k \rangle$.

*Proof.*

$$\langle p(y) \cdot (1+y), \sum_{k=0}^{N-1} \omega_{k+1} y^k \rangle = \sum_{k=0}^{N-2} p_k \omega_{k+1} + \sum_{k=0}^{N-2} p_k \omega_{k+2} = \sum_{k=0}^{N-2} (\omega_{k+1} + \omega_{k+2}) p_k$$

$$= \langle p(y), \sum_{k=0}^{N-2} (\omega_{k+1} + \omega_{k+2}) y^k \rangle.$$

□

**Corollary 1.** *TreeGrad-Shap (Algorithm 7) computes Beta Shapley value with integral parameters using $O(LD)$ time and $O(D^2 + L)$ space. Without vectorization (Algorithm 6), its space complexity is $O(L)$.*

*Proof.* By linearity and (Dubey et al., 1981, Theorem 1(a′)),

$$\phi^{\text{Shap}}(f_{\mathbf{x}}) = \int_0^1 \nabla \overline{f}_{\mathbf{x}}(t \cdot \mathbf{1}_N) \mathrm{d}t = \int_0^1 \sum_{v \in L_r} \nabla \overline{R}_{\mathbf{x}}^v(t \cdot \mathbf{1}_N) \mathrm{d}t$$

Using Eq. (7),

$$\nabla_i \overline{R}_{\mathbf{x}}^v(t \cdot \mathbf{1}_N) = \sum_{S \subseteq [N] \setminus i} t^s (1-t)^{N-1-s} [R_{\mathbf{x}}^v(S \cup i) - R_{\mathbf{x}}^v(S)],$$

which corresponds to the weighted Banzhaf value parameterized by $t$ (Li & Yu, 2023). Suppose $j \notin F_v$, and notice that $R_{\mathbf{x}}^v(S \cup j) = R_{\mathbf{x}}^v(S)$ for every $S \subseteq [N]$. Then,

$$\nabla_i \overline{R}_{\mathbf{x}}^v(t \cdot \mathbf{1}_N) = \sum_{S \subseteq [N] \setminus \{i,j\}} t^s (1-t)^{N-2-s} [R_{\mathbf{x}}^v(S \cup i) - R_{\mathbf{x}}^v(S)].$$

By removing all features not in $F_v$, we have

$$\nabla_i \overline{R}_{\mathbf{x}}^v(t \cdot \mathbf{1}_N) = \sum_{S \subseteq F_v \setminus i} t^s (1-t)^{|F_v|-1-s} [R_{\mathbf{x}}^v(S \cup i) - R_{\mathbf{x}}^v(S)]. \tag{13}$$

Since $|F_v| \leq D$, the degree of the polynomial $\nabla_i \overline{f}_{\mathbf{x}}(t \cdot \mathbf{1}_N) = \sum_{v \in L_r} \nabla_i \overline{R}_{\mathbf{x}}^v(t \cdot \mathbf{1}_N)$ is at most $D-1$. According to the Gauss-Legendre quadrature rule, we have

$$\phi^{\text{Shap}}(f_{\mathbf{x}}) = \sum_{\ell=1}^{\lceil \frac{D}{2} \rceil} \kappa_\ell \cdot \nabla \overline{f}_{\mathbf{x}}(t_\ell \cdot \mathbf{1}_N).$$

According to Hale & Townsend (2013), $\{t_\ell\}_\ell$ and $\{\kappa_\ell\}_\ell$ can be computed in $O(D)$ time.

Since each gradient $\overline{f}_{\mathbf{x}}(t_\ell \cdot \mathbf{1}_N)$ can be computed in $O(L)$ time using $O(L)$ space, the space complexity is reduced by reusing the same allocated space. This proves the validity of TreeGrad-Shap, as shown in Algorithm 6. □

**Proposition 3.** *Suppose the optimal set $S^*$ for the problem (5) is unique, then $\mathbf{1}_{S^*}$ whose $i$-th entry is 1 if $i \in S^*$ and 0 otherwise is the unique optimal solution to the relaxed problem (6).*

*Proof.* For simplicity, we write $U(S) = \frac{1}{2}(f_{\mathbf{x}}(S) - f_{\mathbf{x}}([N] \setminus S))$, and thus $\overline{U}(\mathbf{z}) = \frac{1}{2}(\overline{f}_{\mathbf{x}}(\mathbf{z}) - \overline{f}_{\mathbf{x}}(\mathbf{1}_N - \mathbf{z}))$. Let $U^* = \arg\max_{S \subseteq [N]} U(S)$. For each $\mathbf{z} \notin \{0,1\}^N$, notice that $\overline{U}(\mathbf{z})$ is just a weighted average of $\{U(S)\}_{S \subseteq [N]}$ because

$$\sum_{S \subseteq [N]} \left( \prod_{i \in S} z_i \right) \left( \prod_{i \notin S} (1-z_i) \right) = \prod_{i=1}^N (1 - z_i + z_i) = 1. \tag{14}$$

Therefore, $\overline{U}(\mathbf{z}) \leq U^*$. For the sake of contradiction, suppose $\overline{U}(\mathbf{z}) = U^*$, then there must exist two distinct subsets $S_1$ and $S_2$ such that $U(S_1) = U(S_2) = U^*$, which contradicts the assumption that $S^*$ is unique. Therefore, we have $\overline{f}_{\mathbf{x}}(\mathbf{z}) < f^*$. □

---

**Algorithm 6** TreeGrad-Shap

---

**Input:** Decision tree $\mathcal{T} = (V, E)$ with root $r$ and depth $D$, instance $\mathbf{x} \in \mathbb{R}^N$, integral parameter $(\alpha, \beta)$ that specifies some Beta Shaple value

**Output:** Shapley value $\phi$

90   Initialize $\phi = \mathbf{0}_N$

91   $M \leftarrow \min(D, N) + \alpha + \beta - 2$

92   Generate $\lceil \frac{M}{2} \rceil$ pair of weights and nodes $\{(\kappa_\ell, t_\ell)\}$ according to the Gauss-Legendre quadrature rule          `// on the closed interval [0,1]`

93   **for** $\ell = 1, 2, \ldots, \lceil \frac{M}{2} \rceil$ **do**

94      $\phi \leftarrow \phi + \kappa_\ell \cdot \text{TreeGrad}(\mathcal{T}, \mathbf{x}, t_\ell \cdot \mathbf{1}_N) \cdot t_\ell^{\beta-1}(1 - t_\ell)^{\alpha-1}/B(\alpha, \beta)$.

---

**Algorithm 7** TreeGrad-Shap (Vectorized)

---

**Input:** Decision tree $\mathcal{T} = (V, E)$ with root $r$, instance $\mathbf{x} \in \mathbb{R}^N$, integral parameter $(\alpha, \beta)$ that specifies some Beta Shaple value

**Output:** Shapley value $\phi$

95   **Function** `traverse(`$v, \mathbf{s} = \mathbf{b}$`):`

96     $e \leftarrow (m_v \rightarrow v)$

97     **if** $e^\uparrow$ *exists* **then**

98       $\mathbf{s} \leftarrow \mathbf{s} \cdot (1 - \mathbf{t} + \gamma_e \cdot \mathbf{t})/(1 - \mathbf{t} + \gamma_{e^\uparrow} \cdot \mathbf{t})$          `// element-wise`

99     **else**

100       $\mathbf{s} \leftarrow \mathbf{s} \cdot (1 - \mathbf{t} + \gamma_e \cdot \mathbf{t})$

101     **if** $v$ *is a leaf* **then**

102       $\mathbf{s} \leftarrow \frac{\varrho_v c_v}{c_r} \cdot \mathbf{s}$

103     **else**

104       $\mathbf{s} \leftarrow$ `traverse(`$a_v, \mathbf{s}$`)` $+$ `traverse(`$b_v, \mathbf{s}$`)`

105     **if** $e^\uparrow$ *exists* **then**

106       $H[h_{e^\uparrow}] \leftarrow H[h_{e^\uparrow}] - \mathbf{s}$

107     $H[v] \leftarrow H[v] + \mathbf{s}$

108     $\phi_{l_e} \leftarrow \phi_{l_e} + (\gamma_e - 1) \cdot \langle H[v]/(1 - \mathbf{t} + \gamma_e \cdot \mathbf{t}), \kappa \rangle$

109     **return** $s$

110   $M \leftarrow \lceil \frac{\min(D,N)+\alpha+\beta-2}{2} \rceil$

111   Generate the weight vector $\kappa \in \mathbb{R}^M$ and the node vector $\mathbf{t} \in \mathbb{R}^M$ according to the Gauss-Legendre quadrature rule          `// on the closed interval [0,1]`

112   Compute $\mathbf{b} \in \mathbb{R}^M$ by $b_\ell \leftarrow t_\ell^{\beta-1}(1 - t_\ell)^{\alpha-1}/B(\alpha, \beta)$

113   Initialize $H[v] = \mathbf{0}_M$ for every $v \in V$

114   Initialize $\phi = \mathbf{0}_N$

115   `traverse(`$a_r$`)`

116   `traverse(`$b_r$`)`

117   **return** $\phi$

---

**Lemma 1.** *We have* $\nabla_i \overline{R}_{\mathbf{x}}^v(\mathbf{z}) = (\gamma_{i,v} - 1) \cdot \frac{H_v}{1 - z_i + z_i \gamma_{i,v}}$, *where* $H_v = \frac{\varrho_v c_v}{c_r} \cdot \prod_{j \in F_v}(1 - z_j + z_j \gamma_{j,v})$.

*Proof.* Similar to the derivation of Eq. (13), there is

$$\nabla_i \overline{R}_{\mathbf{x}}^v(\mathbf{z}) = \frac{\varrho_v c_v}{c_r}(\gamma_{i,v} - 1) \sum_{S \subseteq F_v \setminus i} \left( \prod_{j \in S} z_j \gamma_{j,v} \right) \left( \prod_{j \notin S \cup i} (1 - z_j) \right)$$

Notice that $\sum_{S \subseteq F_v \setminus i} \left( \prod_{j \in S} z_j \gamma_{j,v} \right) \left( \prod_{j \notin S \cup i}(1 - z_j) \right) = \prod_{j \in F_v \setminus i}(1 - z_j + z_j \gamma_{j,v})$.      $\square$

**Lemma 2.** $\nabla_i \overline{f}_{\mathbf{x}}(\mathbf{z}) = \sum_{\substack{e \in E \\ l_e = i}} (\gamma_e - 1) \cdot \frac{\sum_{v \in L_{h_e}^\dagger} H_v}{1 - z_i + z_i \gamma_e}$.

---

**Algorithm 8** Computation of $f_{\mathbf{x}}$

---

**Input:** Decision tree $\mathcal{T} = (V, E)$ with root $r$, instance $\mathbf{x} \in \mathbb{R}^N$, subset $S \subseteq [N]$
**Output:** Realization of $\mathbb{E}_{\mathbf{X}_{S^c}}[f(\mathbf{x}_S, \mathbf{X}_{S^c})]$

118 **Function** traverse($v$):
119    **if** $v$ *is a leaf* **then**
120      **return** $\varrho_v$
121    **if** $l_v \in S$ **then**
122      **if** $x_{l_v} \leq \tau_v$ **then**
123        **return** traverse($a_v$)
124      **else**
125        **return** traverse($b_v$)
126    **else**
127      **return** $\frac{c_{a_v}}{c_v}$ traverse($a_v$) $+ \frac{c_{b_v}}{c_v}$ traverse($b_v$)
128 **return** traverse($r$)

---

**Algorithm 9** Efficient Evaluation of $\overline{f}_{\mathbf{x}}(\mathbf{z})$

---

**Input:** Decision tree $\mathcal{T} = (V, E)$ with root $r$, instance $\mathbf{x} \in \mathbb{R}^N$, point $\mathbf{z} \in [0, 1]^N$
**Output:** The evaluation of $\overline{f}_{\mathbf{x}}(\mathbf{z})$

129 **Function** traverse($v$, $s = 1$):
130    $e \leftarrow (m_v \to v)$
131    **if** $e^{\uparrow}$ *exists* **then**
132      $s \leftarrow \frac{s}{1 - z_{l_{e^{\uparrow}}} + z_{l_{e^{\uparrow}}} \gamma_{e^{\uparrow}}} \cdot (1 - z_{l_e} + z_{l_e} \gamma_e)$
133    **else**
134      $s \leftarrow s \cdot (1 - z_{l_e} + z_{l_e} \gamma_e)$
135    **if** $v$ *is a leaf* **then**
136      $s \leftarrow \frac{\varrho_v c_v}{c_r} \cdot s$
137    **else**
138      $s \leftarrow$ traverse($a_v$, $s$) $+$ traverse($b_v$, $s$)
139    **return** $s$
140 **return** traverse($a_r$) $+$ traverse($b_r$)

---

*Proof.* Let $i$ be fixed, one can verify that $\{L_{h_e}^{\dagger}\}_{e \in E, l_e = i}$ forms a partition of $L_r$. Therefore,

$$\nabla_i \overline{f}_{\mathbf{x}}(\mathbf{z}) = \sum_{v \in L_r} (\gamma_{i,v} - 1) \cdot \frac{H_v}{1 - z_i + z_i \gamma_{i,v}} = \sum_{\substack{e \in E \\ l_e = i}} \sum_{v \in L_{h_e}^{\dagger}} (\gamma_{i,v} - 1) \cdot \frac{H_v}{1 - z_i + z_i \gamma_{i,v}}.$$

By the definition of $L_{h_e}^{\dagger}$, we have $\gamma_{i,v} = \gamma_e$, and thus the conclusion follows. □

## G    EXPERIMENTS

The full procedure of TreeGrad is presented in Algorithm 11. The information of the datasets we employ is summarized in Table 1.

**More experimental results.** Using decision trees, additional experimental results are presented in Figures 6 and 7 for classification tasks, whereas results for regression tasks are shown in Figure 8. Our TreeGrad-Ranker also works for gradient boosted trees (Friedman, 2001). We train gradient boosted trees on the same datasets using GradientBoostingClassifier and GradientBoostingRegressor from the scikit-learn library. The corresponding results are presented in Figures 9–13. For classification tasks, the output of each gradient boosted trees model is set to be the logit of the selected class.

**More experimental settings.** For all trained decision trees, the only hyperparameter we control is the tree depth $D$. The random seed of training is set to 2025 to ensure reproducibility, while all

---

**Algorithm 10** TreeGrad-Ranker with ADAM

---

**Input:** Decision tree $\mathcal{T} = (V, E)$ with root $r$, instance $\mathbf{x} \in \mathbb{R}^N$, learning rate $\epsilon > 0$, iteration $T$.
        Default hyperparameters of ADAM are: $\beta_1 = 0.9$, $\beta_2 = 0.999$ and $\epsilon' = 10^{-8}$
**Output:** Vector of feature scores $\boldsymbol{\zeta}$

141  Initialize $\mathbf{z} = 0.5 \cdot \mathbf{1}_N$, $\boldsymbol{\zeta} = \mathbf{0}_N$, $\mathbf{m}_0 = \mathbf{0}_N$ and $\mathbf{v}_0 = \mathbf{0}_N$

142  **for** $t = 1, 2, \ldots, T$ **do**

143      $\mathbf{g} \leftarrow \frac{1}{2}(\text{TreeGrad}(\mathcal{T}, \mathbf{x}, \mathbf{z}) + \text{TreeGrad}(\mathcal{T}, \mathbf{x}, \mathbf{1}_N - \mathbf{z}))$

144      $\boldsymbol{\zeta} \leftarrow \frac{t-1}{t} \cdot \boldsymbol{\zeta} + \frac{1}{t} \cdot \mathbf{g}$

145      $\mathbf{m}_t \leftarrow \beta_1 \cdot \mathbf{m}_{t-1} + (1 - \beta_1) \cdot \mathbf{g}$

146      $\mathbf{v}_t \leftarrow \beta_2 \cdot \mathbf{v}_{t-1} + (1 - \beta_2) \cdot \mathbf{g}^2$

147      $\hat{\mathbf{m}}_t \leftarrow \mathbf{m}_t / (1 - \beta_1^t)$

148      $\hat{\mathbf{v}}_t \leftarrow \mathbf{v}_t / (1 - \beta_2^t)$

149      $\mathbf{z} \leftarrow \mathbf{z} + \epsilon \cdot \hat{\mathbf{m}}_t / \sqrt{\hat{\mathbf{v}}_t + \epsilon'}$

150      $\mathbf{z} \leftarrow \text{Proj}(\mathbf{z})$                          // $z_i \leftarrow \min(\max(0, z_i), 1)$

151  **return** $\boldsymbol{\zeta}$

---

Table 1: Summary of the datasets used in this work.

| Dataset | #Instances | #Features | Link | Task | #Classes |
|---|---|---|---|---|---|
| FOTP (Bridge et al., 2014) | 6,118 | 51 | https://openml.org/d/1475 | classification | 6 |
| GPSP (Madeo et al., 2013) | 9,873 | 32 | https://openml.org/d/4538 | classification | 5 |
| jannis | 83,733 | 54 | https://openml.org/d/41168 | classification | 4 |
| MinibooNE (Roe et al., 2005) | 130,064 | 50 | https://openml.org/d/41150 | classification | 2 |
| philippine | 5,832 | 308 | https://openml.org/d/41145 | classification | 2 |
| spambase | 4,601 | 57 | https://openml.org/d/44 | classification | 2 |
| BT (Kawala et al., 2013) | 583,250 | 77 | https://openml.org/d/4549 | regression | - |
| superconduct | 21,263 | 81 | https://openml.org/d/43174 | regression | - |
| wave_energy | 72,000 | 48 | https://openml.org/d/44975 | regression | - |

other hyperparameters are left at their scikit-learn defaults. The tree depth used for each dataset is as reported in the corresponding figure. For gradient boosted trees, we set the number of individual decision trees to 5 for all datasets. Table 2 summarizes both the sizes of the trained models and their performance on the test datasets.

**Experimental details for producing Figure 2.** For plotting the condition number of TreeShap-K, we set $\gamma = 1$, which appears in Eq. (12). For reporting the inaccuracy when computing the Shapley value, the used decision tree $f$ is trained on the dataset Click_prediction_small with 11 features.[7] With $N = 11$, it is tractable to compute the ground truths using Eq. (1). While varying the depth $D$, we report the average error $\|\hat{\phi} - \phi\|$ over 5 different instances of $\mathbf{x}$. The algorithm for running Linear TreeShap is from the public repository provided by Yu et al. (2022).[8] For a fair comparison, the inaccuracy of Linear TreeShap (well-conditioned) is obtained by simply replacing the Vandermonde matrix used in Linear TreeShap with the well-conditioned one.

**Further inaccuracy analysis.** In Algorithm 3, the number of nodes is determined by $M \leftarrow \min(D, N)$. If we include TreeProb in Figure 2, $M$ would be fixed as 11, since $N = 11$. Consequently, this setting would not reveal when TreeProb becomes numerically unstable. For completeness, we therefore also include TreeProb and TreeGrad-Shap by setting $M \leftarrow D$. In addition, we also compare against the path-dependent TreeShap (Lundberg et al., 2020) implemented in the SHAP package.[9] The corresponding results are shown in Figure 2. Specially, TreeProb (modified) refers to the variant that adopts the same numerical stabilization trick as Linear TreeShap, as described in Appendix B. Clearly, our TreeGrad-Shap is the most numerically stable algorithm, which further implies that TreeGrad itself is numerically stable. The numerical error of TreeProb is already $10^{-8}$ when $M = 35$, and increases to 10 when $M = 65$.

---

[7] https://openml.org/d/1219.

[8] https://github.com/yupbank/linear_tree_shap/blob/main/linear_tree_shap/fast_linear_tree_shap.py.

[9] https://github.com/shap/shap.

Table 2: The sizes of the trained tree models and their performance on the test datasets. We report test accuracy for classification tasks and the coefficient of determination $R^2$ for regression tasks.

| Dataset | Decision Tree | | | | Gradient Boosted Trees | | | |
|---|---|---|---|---|---|---|---|---|
| | $R^2$/Acc | Depth | #Leaves | #Nodes | $R^2$/Acc | Depth | #Leaves | #Nodes |
| FOTP | 0.537 | 20 | 1,275 | 2,549 | 0.568 | 20 | 43,103 | 86,176 |
| GPSP | 0.498 | 10 | 481 | 961 | 0.572 | 10 | 11,195 | 22,365 |
| jannis | 0.588 | 40 | 11,050 | 22,099 | 0.613 | 40 | 365,809 | 731,598 |
| MinibooNE | 0.899 | 20 | 4,108 | 8,215 | 0.894 | 20 | 25,924 | 51,843 |
| philippine | 0.709 | 15 | 372 | 743 | 0.724 | 15 | 2,572 | 5,139 |
| spambase | 0.925 | 15 | 166 | 331 | 0.922 | 15 | 977 | 1,949 |
| BT | 0.901 | 50 | 408,461 | 816,921 | 0.609 | 50 | 2,042,440 | 4,084,875 |
| superconduct | 0.863 | 10 | 697 | 1,393 | 0.578 | 10 | 3,594 | 7,183 |
| wave_energy | 0.525 | 30 | 56,814 | 113,627 | 0.465 | 30 | 284,265 | 568,525 |

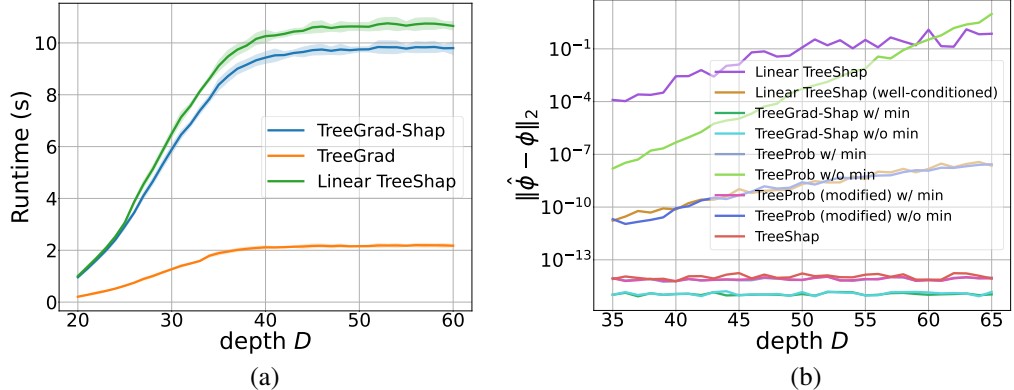

Figure 5: (a) The runtime of Linear TreeShap and our TreeGrad-Shap (Algorithm 7) for computing the Shapley value, and of our TreeGrad (Algorithm 1) for computing the Banzhaf value. (b) The numerical errors of different algorithms.

**Runtime analysis.** We conduct experiments to compare the runtime of Linear TreeShap and our TreeGrad-Shap (Algorithm 7) for computing the Shapley value, and of our TreeGrad (Algorithm 1) for computing the Banzhaf value. Decision trees trained on the BT dataset are employed, as their sizes are the largest according to Table 2. The result is presented in Figure 5. Each curve is averaged over 200 instances, with shaded areas representing the corresponding standard deviation. In theory, the time complexities of Linear TreeShap and our TreeGrad-Shap are equally efficient. However, as shown in Figure 2, the numerical error of Linear TreeShap can be up to $10^{15}$ times larger than that of our TreeGrad-Shap. In contrast, the time complexity of our TreeGrad is $O(L)$.

**How to select $T$ and $\epsilon$ in TreeGrad-Ranker.** The learning rate $\epsilon$ is crucial for the convergence of maximizing the objective in Eq. (6). In particular, each $\overline{f}_{\mathbf{x}}(\mathbf{z})$ can be efficiently evaluated using Algorithm 9, which is derived from our TreeGrad (Algorithm 1) by removing the traversing-up procedure. Therefore, the choice of $\epsilon$ can be guided by observing the changes in $\frac{1}{2}(\overline{f}_{\mathbf{x}}(\mathbf{z}) - \overline{f}_{\mathbf{x}}(\mathbf{1}_N - \mathbf{z}))$ during gradient ascent. Specifically, we select the largest $\epsilon$ in the candidate set $\{\ldots, 0.1, 0.5, 1, 5, 10, \cdots\}$ the provides consistent convergence in $\frac{1}{2}(\overline{f}_{\mathbf{x}}(\mathbf{z}) - \overline{f}_{\mathbf{x}}(\mathbf{1}_N - \mathbf{z}))$, which gives $\epsilon = 5$. Next, using decision trees, we present the performance across different combinations of $T$ and $\epsilon$. The results obtained with gradient ascent are shown in Figures 14–18, whereas those for ADAM are presented in Figures 19–23. Our observations are: (i) the resulting performance improves and converges as $T$ increases; (ii) TreeGrad-Ranker with ADAM converges earlier than gradient ascent, as performance with ADAM reaches near convergence across all the used datasets at $T = 10$; (iii) lower $\epsilon$ does not improve performance.

---

**Algorithm 11** TreeGrad (Full Procedure)

---

**Input:** Decision tree $\mathcal{T} = (V, E)$ with root $r$, instance $\mathbf{x} \in \mathbb{R}^N$, point $\mathbf{z} \in [0,1]^N$
**Output:** Gradient $\mathbf{g}$

152 **Function** `traverse`$(v, s = 1)$:
153    $e \leftarrow (m_v \rightarrow v)$
154    **if** $1 - z_{l_e} + z_{l_e}\gamma_e = 0$ **then**
155      **if** $e^\uparrow$ *exists* **then**
156        $s \leftarrow \frac{s}{1 - z_{l_{e^\uparrow}} + z_{l_{e^\uparrow}}\gamma_{e^\uparrow}}$
157      **if** $v$ *is a leaf* **then**
158        $s \leftarrow \frac{\varrho_v c_v}{c_r} \cdot s$
159      **else**
160        $s \leftarrow$ `traverse-zero`$(a_v, s, l_e)$ $+$ `traverse-zero`$(b_v, s, l_e)$
161      $g_{l_e} \leftarrow g_{l_e} - s$
162      **return** $0$
163    **else**
164      **if** $e^\uparrow$ *exists* **then**
165        $s \leftarrow \frac{s}{1 - z_{l_{e^\uparrow}} + z_{l_{e^\uparrow}}\gamma_{e^\uparrow}} \cdot (1 - z_{l_e} + z_{l_e}\gamma_e)$
166      **else**
167        $s \leftarrow s \cdot (1 - z_{l_e} + z_{l_e}\gamma_e)$
168      **if** $v$ *is a leaf* **then**
169        $s \leftarrow \frac{\varrho_v c_v}{c_r} \cdot s$
170      **else**
171        $s \leftarrow$ `traverse`$(a_v, s)$ $+$ `traverse`$(b_v, s)$
172      **if** $e^\uparrow$ *exists* **then**
173        $H[h_{e^\uparrow}] \leftarrow H[h_{e^\uparrow}] - s$
174      $H[v] \leftarrow H[v] + s$
175      $g_{l_e} \leftarrow g_{l_e} + (\gamma_e - 1) \cdot \frac{H[v]}{1 - z_{l_e} + z_{l_e}\gamma_e}$
176      **return** $s$

177 **Function** `traverse-zero`$(v, s, l)$:
178    $e \leftarrow (m_v \rightarrow v)$
179    **if** $l_e \neq l$ **then**
180      **if** $1 - z_{l_e} + z_{l_e}\gamma_e = 0$ **then**
181        **return** $0$
182      **if** $e^\uparrow$ *exists* **then**
183        $s \leftarrow \frac{s}{1 - z_{l_{e^\uparrow}} + z_{l_{e^\uparrow}}\gamma_{e^\uparrow}} \cdot (1 - z_{l_e} + z_{l_e}\gamma_e)$
184      **else**
185        $s \leftarrow s \cdot (1 - z_{l_e} + z_{l_e}\gamma_e)$
186    **if** $v$ *is a leaf* **then**
187      $s \leftarrow \frac{\varrho_v c_v}{c_r} \cdot s$
188    **else**
189      $s \leftarrow$ `traverse-zero`$(a_v, s, l)$ $+$ `traverse-zero`$(b_v, s, l)$
190    **return** $s$

191 Initialize $H[v] = 0$ for every $v \in V$
192 Initialize $\mathbf{g} = \mathbf{0}_N$
193 `traverse`$(a_r)$
194 `traverse`$(b_r)$
195 **return** $\mathbf{g}$

---

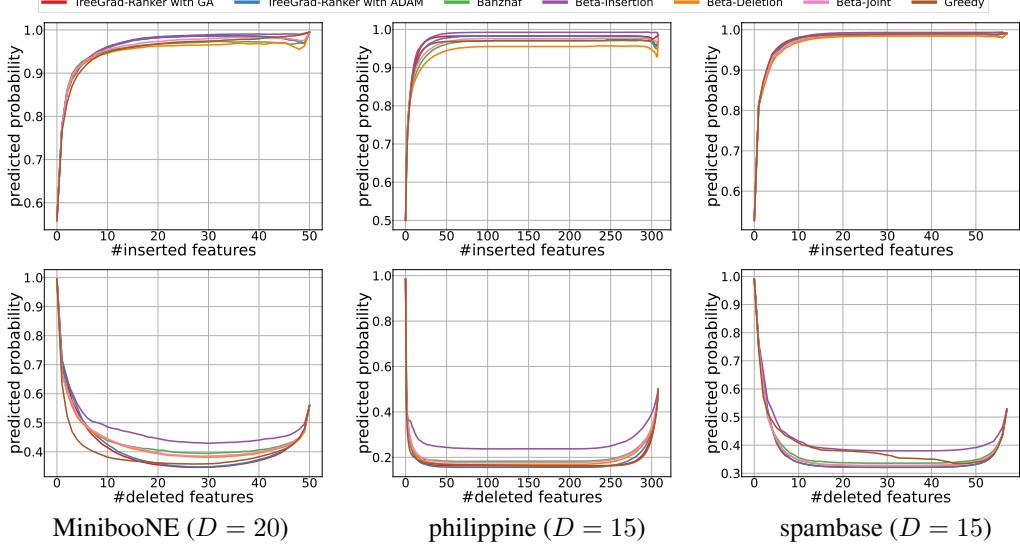

Figure 6: The trained models are **decision trees**. The first row shows the insertion curves, while the second presents the deletion curves. A higher insertion curve or a lower deletion curve indicates better performance. For our TreeGrad-Ranker, we set $T = 100$ and $\epsilon = 5$. Beta-Insertion selects the candidate Beta Shapley value that maximizes the Ins metric for each $f_\mathbf{x}$, whereas Beta-Deletion minimizes the Del metric. Beta-Joint selects the candidate that maximizes the joint metric Ins $-$ Del. The output of each $f(\mathbf{x})$ is set to be the probability of **its predicted class**.

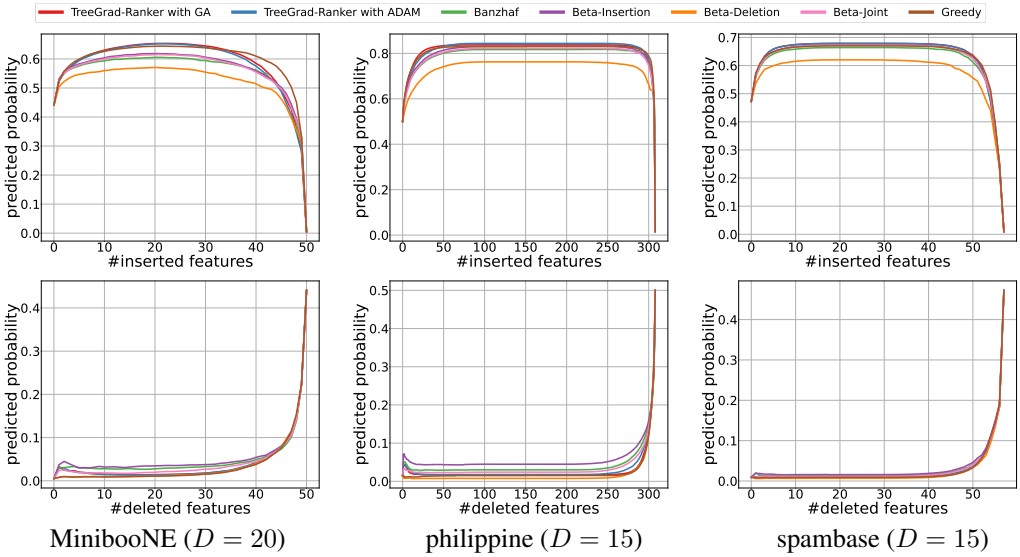

Figure 7: The trained models are **decision trees**. The first row shows the insertion curves, while the second presents the deletion curves. A higher insertion curve or a lower deletion curve indicates better performance. For our TreeGrad-Ranker, we set $T = 100$ and $\epsilon = 5$. Beta-Insertion selects the candidate Beta Shapley value that maximizes the Ins metric for each $f_\mathbf{x}$, whereas Beta-Deletion minimizes the Del metric. Beta-Joint selects the candidate that maximizes the joint metric Ins $-$ Del. The output of each $f(\mathbf{x})$ is set to be the probability of **a randomly sampled non-predicted class**.

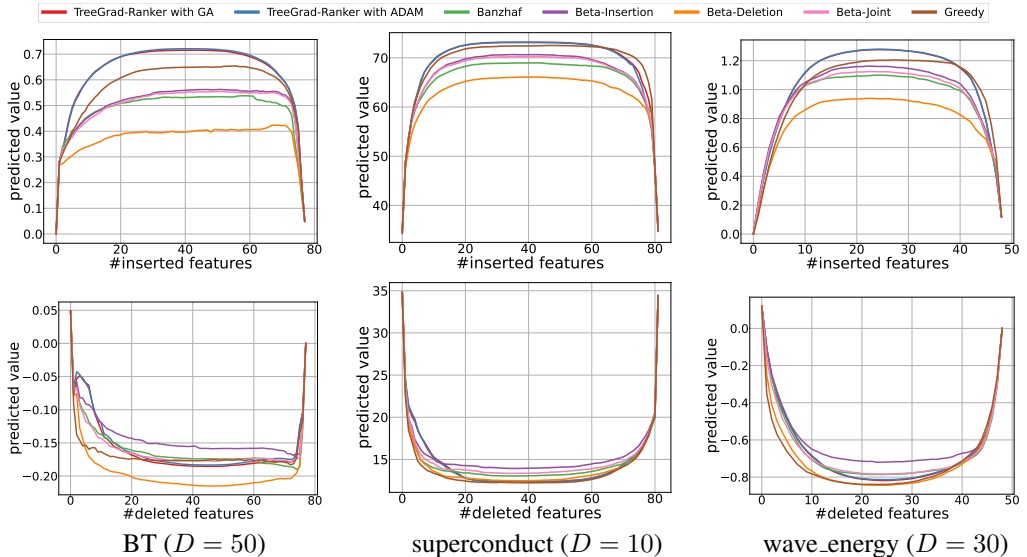

Figure 8: The trained models are **decision trees**. The first row shows the insertion curves, while the second presents the deletion curves. A higher insertion curve or a lower deletion curve indicates better performance. For our TreeGrad-Ranker, we set $T = 100$ and $\epsilon = 5$. Beta-Insertion selects the candidate Beta Shapley value that maximizes the Ins metric for each $f_{\mathbf{x}}$, whereas Beta-Deletion minimizes the Del metric. Beta-Joint selects the candidate that maximizes the joint metric Ins$-$Del. The datasets used are regression tasks.

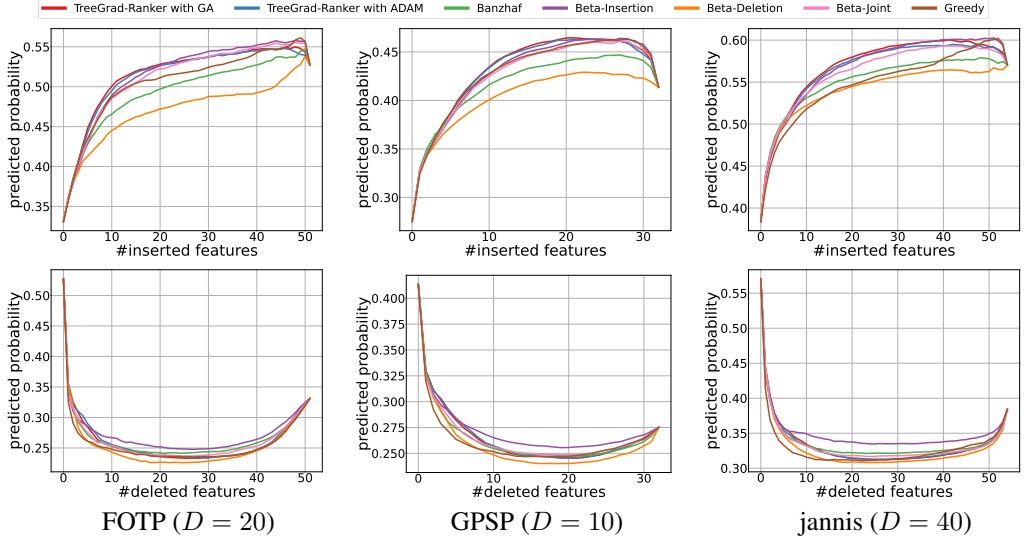

Figure 9: All trained models are **gradient boosted trees** consisting of 5 individual decision trees. The first row shows the insertion curves, while the second presents the deletion curves. A higher insertion curve or a lower deletion curve indicates better performance. For our TreeGrad-Ranker, we set $T = 100$ and $\epsilon = 5$. Beta-Insertion selects the candidate Beta Shapley value that maximizes the Ins metric for each $f_{\mathbf{x}}$, whereas Beta-Deletion minimizes the Del metric. Beta-Joint selects the candidate that maximizes the joint metric Ins $-$ Del. The output of each $f(\mathbf{x})$ is set to be the logit of **its predicted class**.

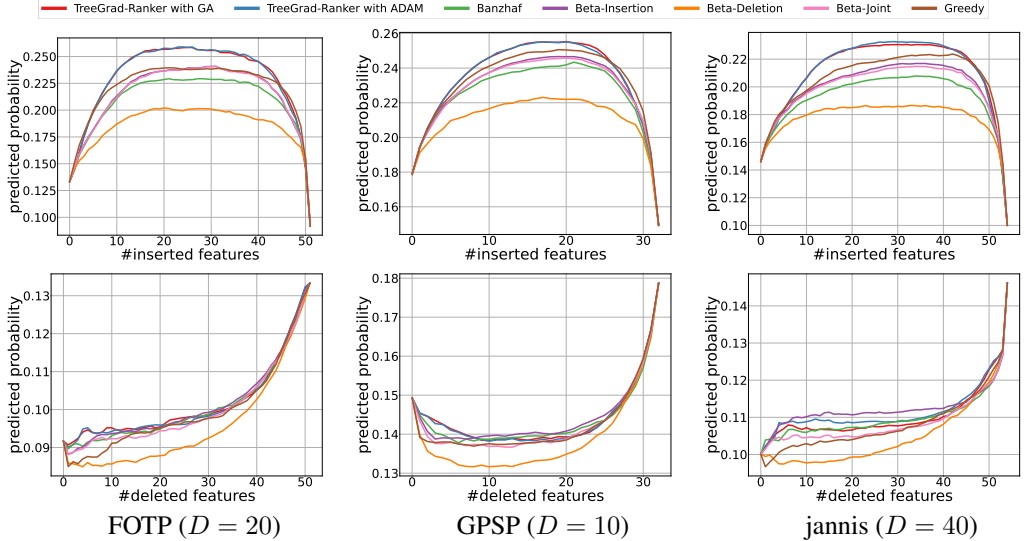

Figure 10: All trained models are **gradient boosted trees** consisting of 5 individual decision trees. The first row shows the insertion curves, while the second presents the deletion curves. A higher insertion curve or a lower deletion curve indicates better performance. For our TreeGrad-Ranker, we set $T = 100$ and $\epsilon = 5$. Beta-Insertion selects the candidate Beta Shapley value that maximizes the Ins metric for each $f_{\mathbf{x}}$, whereas Beta-Deletion minimizes the Del metric. Beta-Joint selects the candidate that maximizes the joint metric Ins $-$ Del. The output of each $f(\mathbf{x})$ is set to be the logit of **a randomly sampled non-predicted class**.

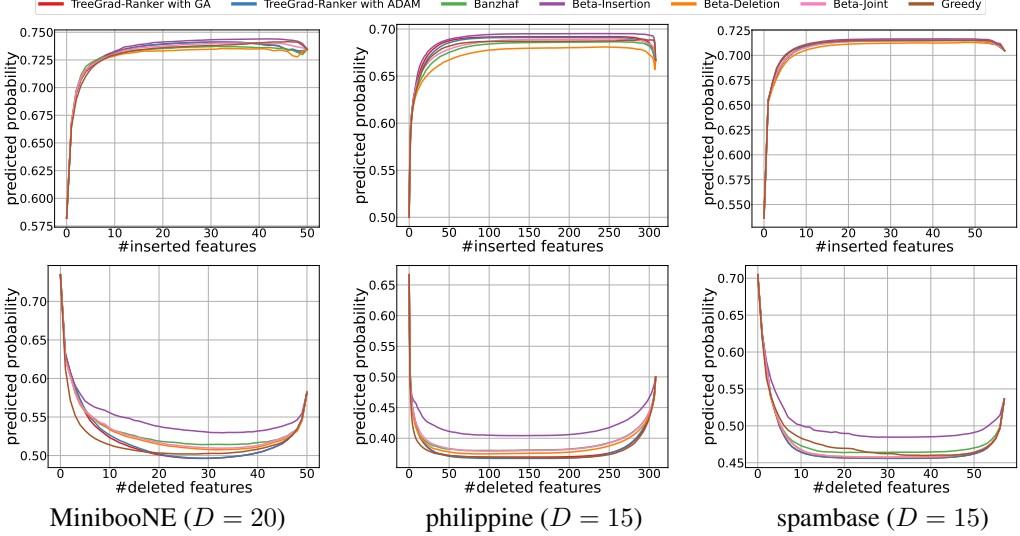

Figure 11: All trained models are **gradient boosted trees** consisting of 5 individual decision trees. The first row shows the insertion curves, while the second presents the deletion curves. A higher insertion curve or a lower deletion curve indicates better performance. For our TreeGrad-Ranker, we set $T = 100$ and $\epsilon = 5$. Beta-Insertion selects the candidate Beta Shapley value that maximizes the Ins metric for each $f_{\mathbf{x}}$, whereas Beta-Deletion minimizes the Del metric. Beta-Joint selects the candidate that maximizes the joint metric Ins $-$ Del. The output of each $f(\mathbf{x})$ is set to be the logit of **its predicted class**.

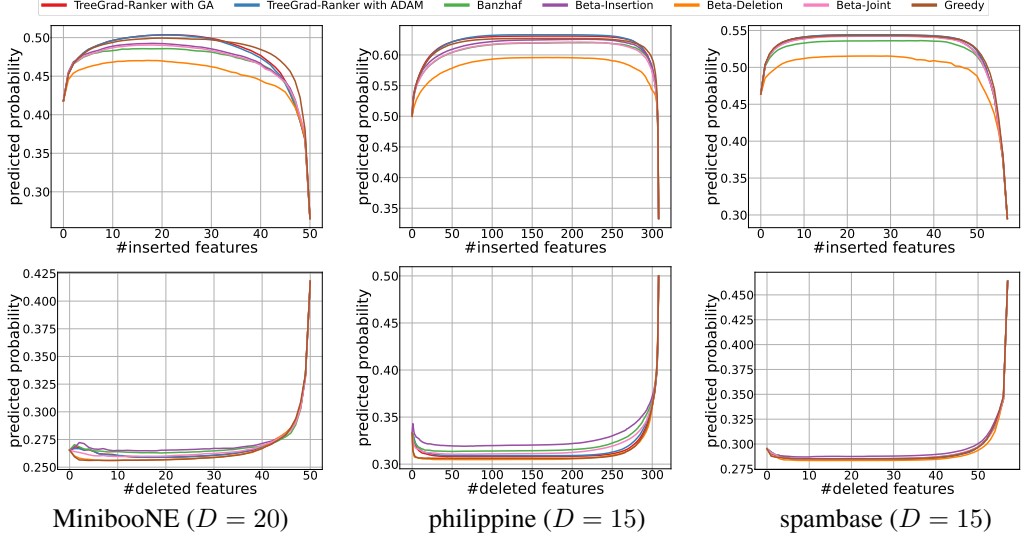

Figure 12: All trained models are **gradient boosted trees** consisting of 5 individual decision trees. The first row shows the insertion curves, while the second presents the deletion curves. A higher insertion curve or a lower deletion curve indicates better performance. For our TreeGrad-Ranker, we set $T = 100$ and $\epsilon = 5$. Beta-Insertion selects the candidate Beta Shapley value that maximizes the Ins metric for each $f_\mathbf{x}$, whereas Beta-Deletion minimizes the Del metric. Beta-Joint selects the candidate that maximizes the joint metric $\text{Ins} - \text{Del}$. The output of each $f(\mathbf{x})$ is set to be the logit of **a randomly sampled non-predicted class**.

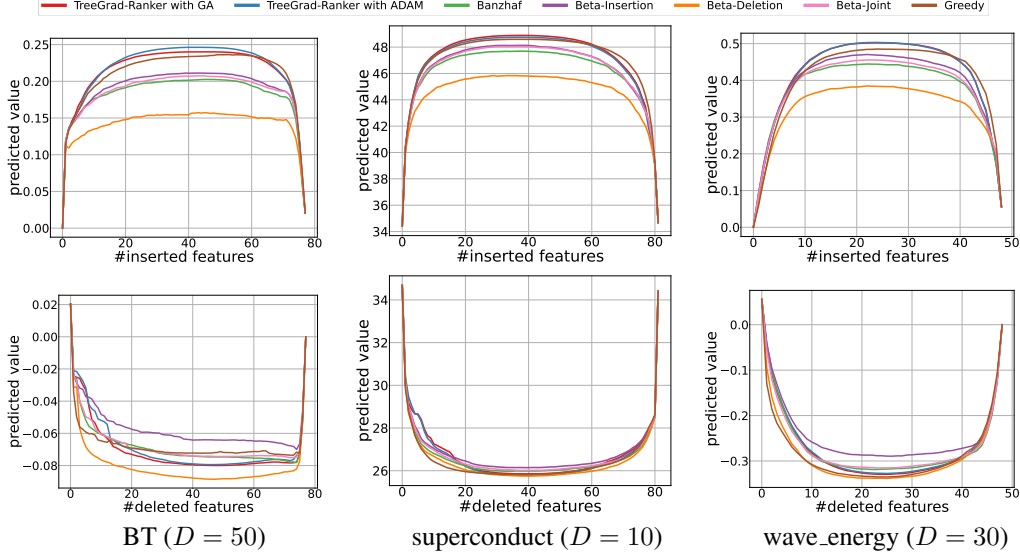

Figure 13: All trained models are **gradient boosted trees** consisting of 5 individual decision trees. The first row shows the insertion curves, while the second presents the deletion curves. A higher insertion curve or a lower deletion curve indicates better performance. For our TreeGrad-Ranker, we set $T = 100$ and $\epsilon = 5$. Beta-Insertion selects the candidate Beta Shapley value that maximizes the Ins metric for each $f_\mathbf{x}$, whereas Beta-Deletion minimizes the Del metric. Beta-Joint selects the candidate that maximizes the joint metric $\text{Ins} - \text{Del}$. The datasets used are regression tasks.

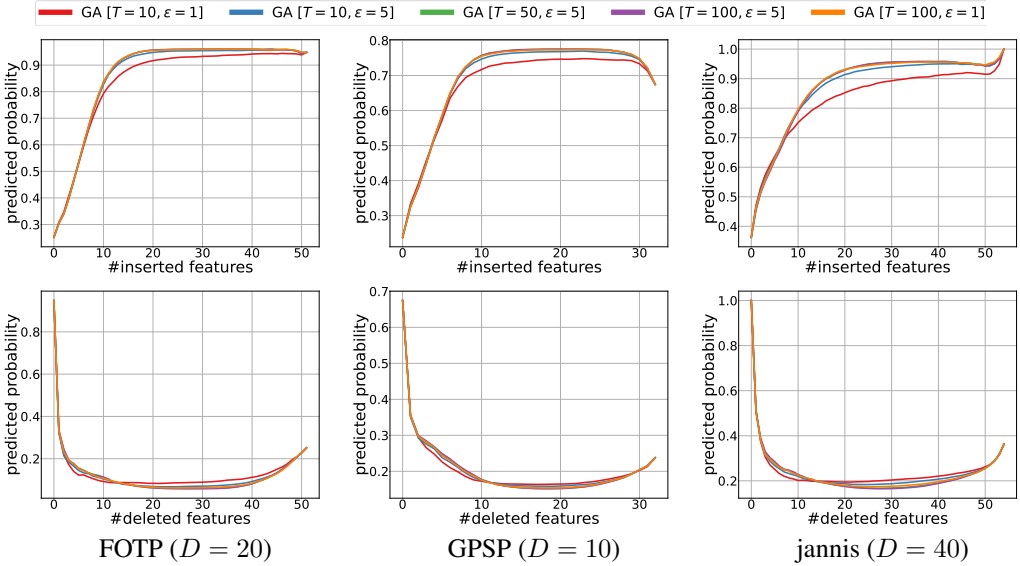

Figure 14: Results of TreeGrad-Ranker with **GA**. All trained models are **decision trees**. The first row shows the insertion curves, while the second presents the deletion curves. A higher insertion curve or a lower deletion curve indicates better performance. The output of each $f(\mathbf{x})$ is set to be the probability of **its predicted class**.

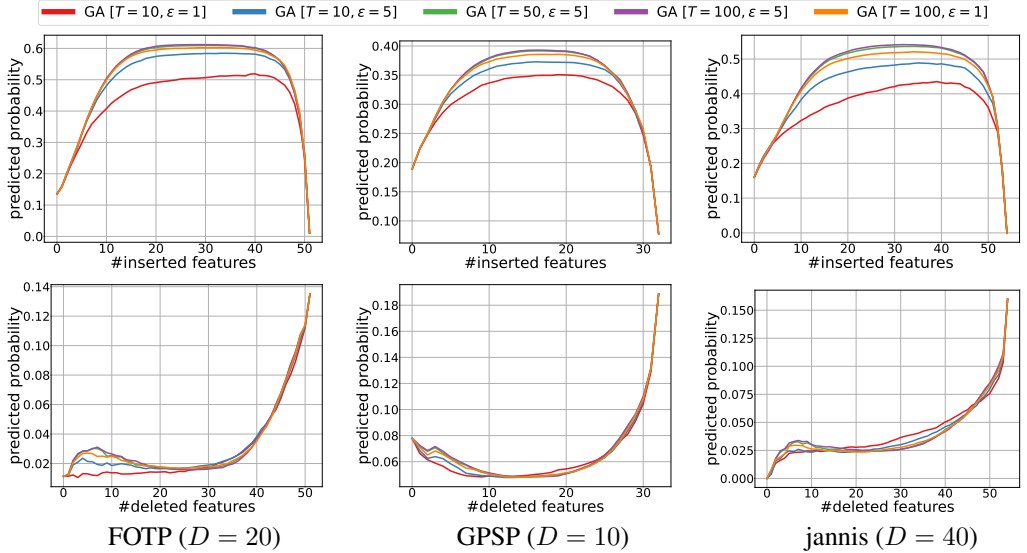

Figure 15: Results of TreeGrad-Ranker with **GA**. All trained models are **decision trees**. The first row shows the insertion curves, while the second presents the deletion curves. A higher insertion curve or a lower deletion curve indicates better performance. The output of each $f(\mathbf{x})$ is set to be the probability of **a randomly sampled non-predicted class**.

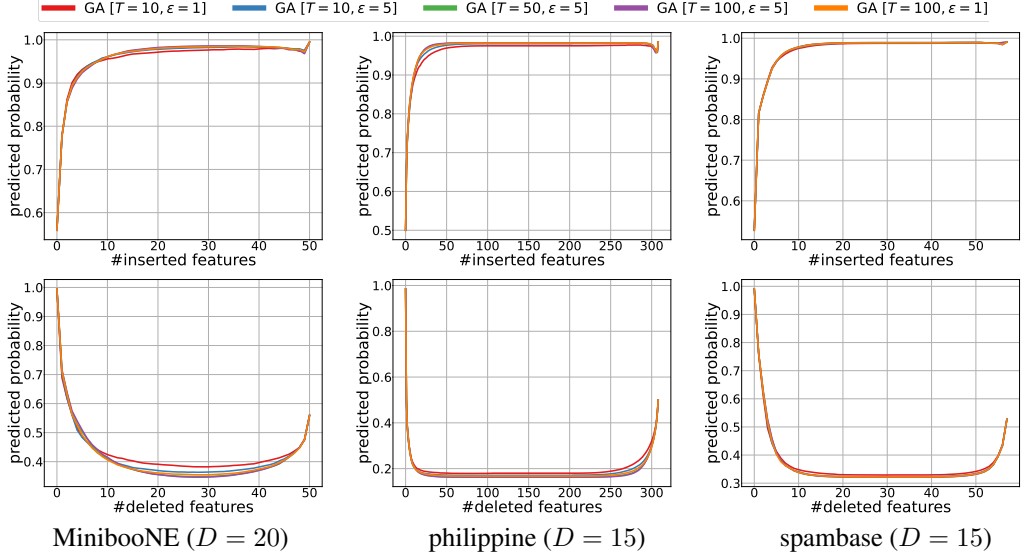

Figure 16: Results of TreeGrad-Ranker with **GA**. All trained models are **decision trees**. The first row shows the insertion curves, while the second presents the deletion curves. A higher insertion curve or a lower deletion curve indicates better performance. The output of each $f(\mathbf{x})$ is set to be the probability of **its predicted class**.

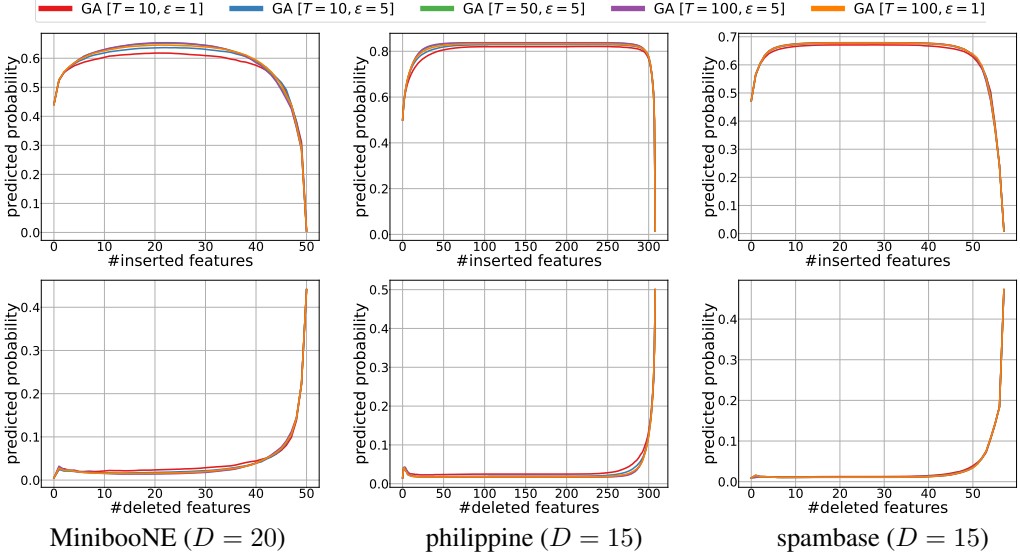

Figure 17: Results of TreeGrad-Ranker with **GA**. All trained models are **decision trees**. The first row shows the insertion curves, while the second presents the deletion curves. A higher insertion curve or a lower deletion curve indicates better performance. The output of each $f(\mathbf{x})$ is set to be the probability of **a randomly sampled non-predicted class**.

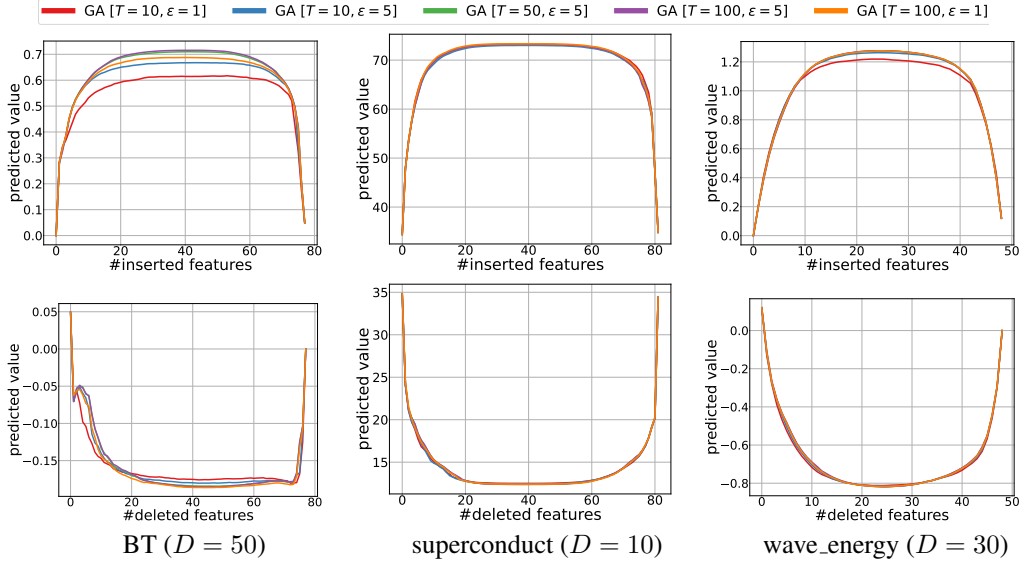

Figure 18: Results of TreeGrad-Ranker with **GA**. All trained models are **decision trees**. The first row shows the insertion curves, while the second presents the deletion curves. A higher insertion curve or a lower deletion curve indicates better performance. The datasets used are regression tasks.

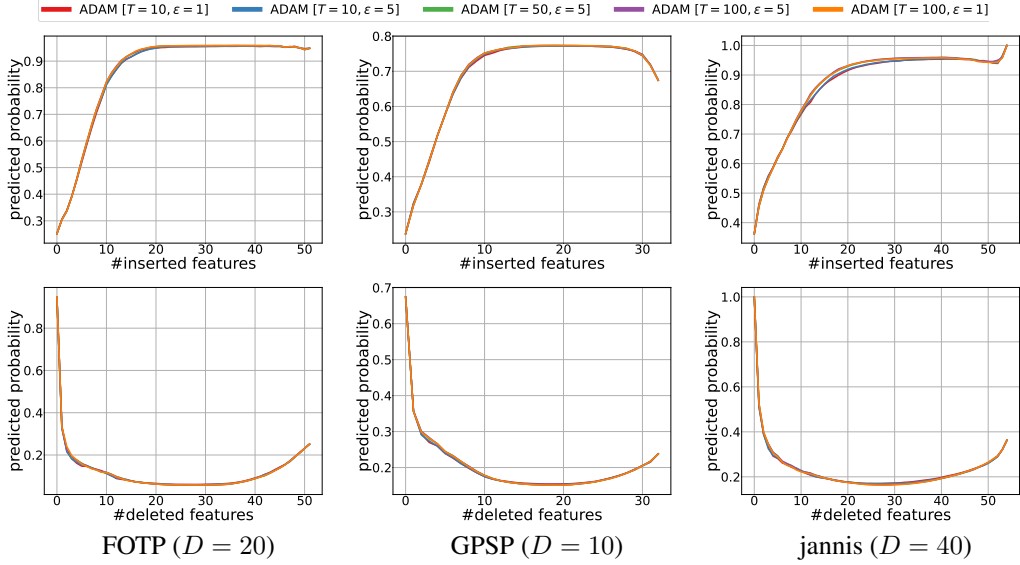

Figure 19: Results of TreeGrad-Ranker with **ADAM**. All trained models are **decision trees**. The first row shows the insertion curves, while the second presents the deletion curves. A higher insertion curve or a lower deletion curve indicates better performance. The output of each $f(\mathbf{x})$ is set to be the probability of **its predicted class**.

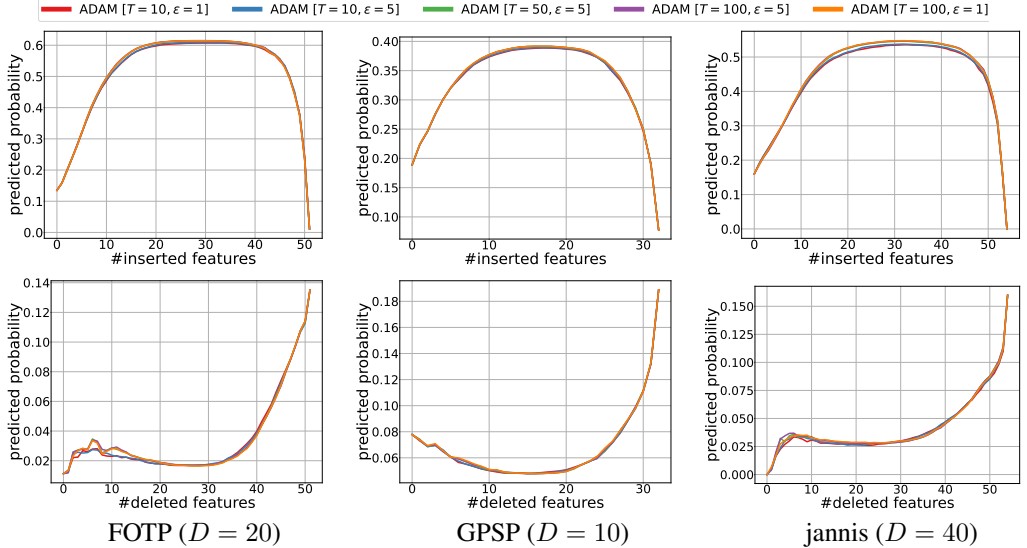

Figure 20: Results of TreeGrad-Ranker with **ADAM**. All trained models are **decision trees**. The first row shows the insertion curves, while the second presents the deletion curves. A higher insertion curve or a lower deletion curve indicates better performance. The output of each $f(\mathbf{x})$ is set to be the probability of **a randomly sampled non-predicted class**.

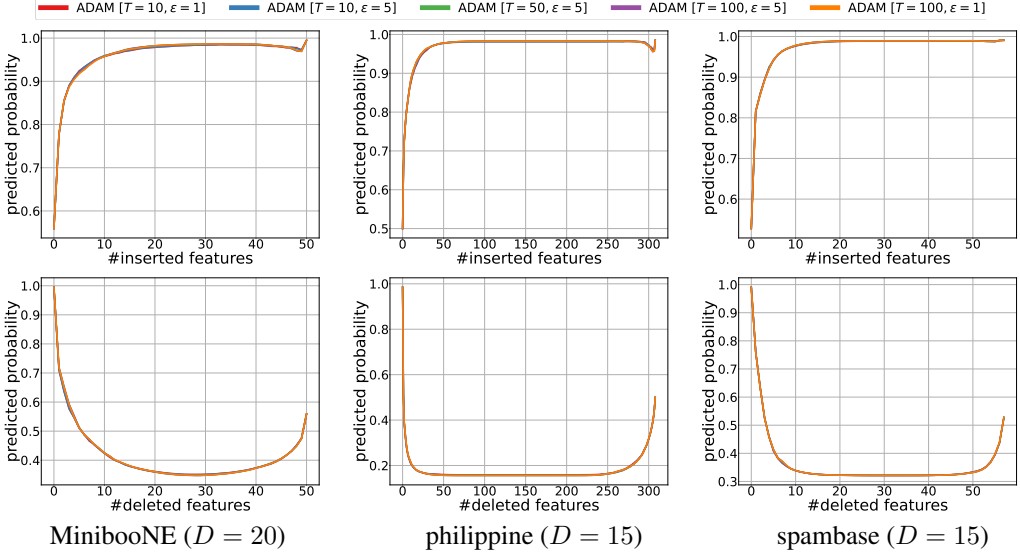

Figure 21: Results of TreeGrad-Ranker with **ADAM**. All trained models are **decision trees**. The first row shows the insertion curves, while the second presents the deletion curves. A higher insertion curve or a lower deletion curve indicates better performance. The output of each $f(\mathbf{x})$ is set to be the probability of **its predicted class**.

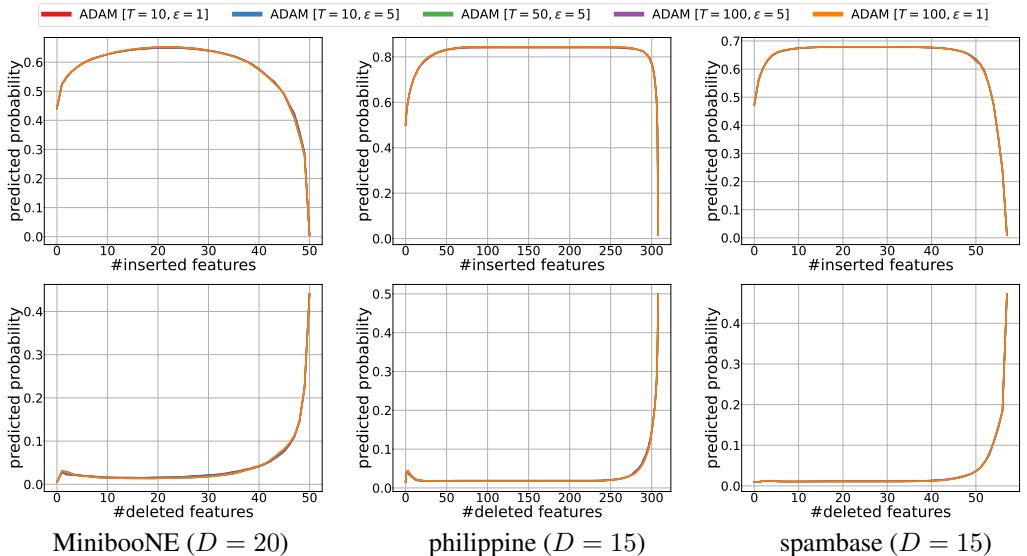

Figure 22: Results of TreeGrad-Ranker with **ADAM**. All trained models are **decision trees**. The first row shows the insertion curves, while the second presents the deletion curves. A higher insertion curve or a lower deletion curve indicates better performance. The output of each $f(\mathbf{x})$ is set to be the probability of **a randomly sampled non-predicted class**.

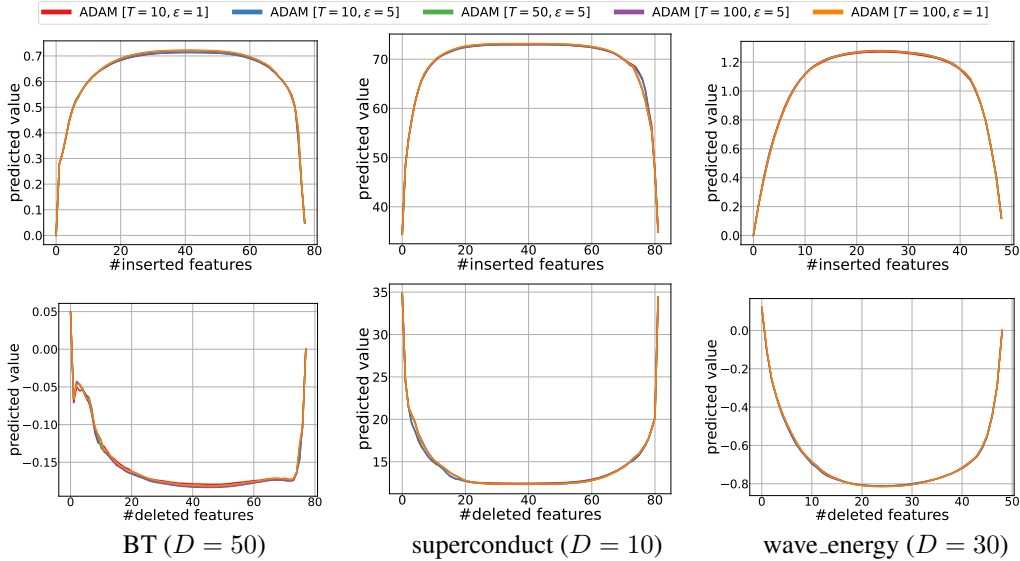

Figure 23: Results of TreeGrad-Ranker with **ADAM**. All trained models are **decision trees**. The first row shows the insertion curves, while the second presents the deletion curves. A higher insertion curve or a lower deletion curve indicates better performance. The datasets used are regression tasks.

