# OpenReview forum: "TreeGrad-Ranker: Feature Ranking via $O(L)$-Time Gradients for Decision Trees"
_ICLR.cc/2026/Conference — ICLR 2026 Poster_

### Official Review · Reviewer_vun4 · 2025-10-28

**Soundness:** 3
**Presentation:** 3
**Contribution:** 3
**Rating:** 6
**Confidence:** 3

**Summary:**

The authors proposed TreeGrad, a way of computing feature importance via solving the multilinear extension of the joint optimization problem. They showed its theoretical guarantees and empirical effectiveness, comparing against other insertion / deletion based methods.
The authors also presented an improved algorithm for computing probabilistic values in decision trees (called TreeProb), and a more numerically stable variant of TreeShap (they call it TreeGrad-Shap).

**Strengths:**

1. The theoretical analysis is thorough, especially the arguments in section 3, optimizing the joint makes a lot of sense
2. Clear explanation of the algorithms, algorithms mentioned are listed clearly in the paper
3. Strong experimental results, empirical evidence supports the authors' claims of their algorithms' effectiveness

**Weaknesses:**

1. Lack of a empirical run-time analysis: the authors claim a O(L) gradient-based method, when comparing with previous approaches with O(LD) complexity. However it is unclear how many gradient steps are enough (10? 100? from figure 3). I think it will be more beneficial to include an empirical run-time analysis.
2. Does randomness affect the method? Since it's gradient based, does different random seed affect the model explanation? I will suggest some analysis around that as well.
3. Also evaluation dataset seems to be of small-scale (9-datasets)

**Questions:**

Minor suggestions:
1. Line 82: ... decision trees serves as the backbone -> ... decision trees serve as the backbone
2. The algorithm names are confusing, TreeGrad, TreeProb, TreeGrad-Shap, TreeGrad-Ranker
3. Main figure 4 could improve its look, the contrast is not clear when it comes the the paper's method / baselines, especially in the bottom parts.

---

> ### Author Response · Authors · 2025-11-25
>
> We are grateful to the reviewer for the insightful feedback and suggestions. We have summarized our updates in our global reponse and address the concerns below.
>
> Q1: Lack of a empirical run-time analysis: the authors claim a O(L) gradient-based method, when comparing with previous approaches with O(LD) complexity. However it is unclear how many gradient steps are enough (10? 100? from figure 3). I think it will be more beneficial to include an empirical run-time analysis.
>
> A1:
> - In Appendix G, we added an experiment comparing the runtime of Linear TreeShap and our TreeGrad-Shap (Algorithm 7) for computing the Shapley value, as well as our TreeGrad (Algorithm 1) for computing the Banzhaf value. The results are shown in Figure 5. In particular, TreeGrad-Shap is slightly faster than Linear TreeShap. Moreover, as shown in Figure 2, the numerical error of Linear TreeShap could be up to $10^{15}$ times larger than that of our TreeGrad-Shap.
>
> - In Appendix G, we also added an experiment for comparing the performance for different combinations of $T$ and $\epsilon$, the results of which are shown in Figures 14-23. We observe that, with $\epsilon=5$, TreeGrad-Ranker with ADAM converges earlier than gradient ascent, as performance with ADAM reaches near convergence across all the used datasets at $T=10$.
>
> Q2: Does randomness affect the method? Since it's gradient based, does different random seed affect the model explanation? I will suggest some analysis around that as well.
>
> A2: Thank you for this question.
> - Our Algorithm 2 does not contain any randomness, because for decision tree we showed that each gradient $\nabla \overline{f}_{\mathbf{x}}$ can be exactly computed in $O(L)$ time using Algorithm 1. Note that the initializer in Algorithm 2 is also fixed.
> - If $f$ is some neural network, then Algorithm 2 would include randomness as each gradient $\nabla \overline{f}_{\mathbf{x}}$ can only be approximated. However, this lies beyond the scope of this submission.
>
>
> Q3: Also evaluation dataset seems to be of small-scale (9-datasets)
>
> A3: Compared to previous related studies, the scale of our employed datasets is larger. For example, in Muschalik et al. (2024, Table 2), the largest numbers of instances and features are 45,222 and 20, respectively. By contrast, in our employed datasets, these numbers are 583,250 and 308, and the smallest number of features is 32.
>
>
> Q4: The algorithm names are confusing, TreeGrad, TreeProb, TreeGrad-Shap, TreeGrad-Ranker
>
> A4: We wish to clarify:
> - TreeGrad (Algorithm) is proposed to compute the **Grad**ients of $\overline{f}_{\mathbf{x}}$ where f is a decision **Tree**.
> - Built upon TreeGrad, TreeGrad-Ranker (Algorithm 2) is proposed to produce effective feature **Rank**ings by jointly optimizing the insertion and deletion metrics.
> - Built upon TreeGrad, TreeGrad-Shap (Algorithm 7) is proposed to accurately compute the **Shap**ley vlaue.
> - TreeProb (Algorithm 4) is proosed to compute **Prob**abilistic values of $f_{\mathbf{x}}$ in polynomial time, where $f$ is a decision **Tree**.
>
>
> Q5: Main figure 4 could improve its look, the contrast is not clear when it comes the the paper's method / baselines, especially in the bottom parts.
>
> A5: Thank you for this suggestion. We have tried to alleviate this issue by only reporting the Beta Shapley value that either achieves the best (i) $\mathrm{Ins}(\pi; f_{\mathbf{x}})$, (ii) $\mathrm{Del}(\pi; f_{\mathbf{x}})$, or (iii) $\mathrm{Ins}(\pi; f_{\mathbf{x}})-\mathrm{Del}(\pi; f_{\mathbf{x}})$.
>
>
> Q6: Typos in line 82
>
> A6: Thank you! We have corrected it.

---

### Official Review · Reviewer_WPUD · 2025-10-31

**Soundness:** 2
**Presentation:** 2
**Contribution:** 2
**Rating:** 4
**Confidence:** 3

**Summary:**

This paper aims to address the reliability issue of probabilistic values of Shapley values and Banzhaf values. The authors proposed TreeGrad-Ranker, which leverages averaged gradients of the multilinear extension as a remedy for decision tree models. The method extends Linear TreeSHAP and achieves fast O(L) time calculation. It also outperforms existing approaches in terms of insertion and deletion metrics on nine empirical dataset.

**Strengths:**

1. The paper is rigorous in its definitions and provides solid theoretical justifications for propose work.
2. The paper identifies the pitfalls of using probabilistic values and proposed an alternative optimization scheme.
3. TreeGrad-Ranker empirically outperform Shapley values, Banzhaf values, Beta-Shapley in terms of insertion and deletion metrics on nine datasets.

**Weaknesses:**

1. The abstract introduces several newly proposed names (TreeGrad, TreeGrad-Ranker, TreeProb and TreeGrad-Shap), which makes it difficult to identify the central contribution. It would be clearer and more engaging if the abstract can focus more on the main problem the paper aims to solve.
2. It’s very difficult to distinguish the performance of different methods in Figure 3 based on solely color. I recommend the authors use additional visual cues, such as distinct shapes, line types for different groups of variables, or just numerical values to demonstrate the proposed methods.
3. Due to the lack of linearity for TreeGrad-Ranker, generalization to popular ensemble trees methods like random forests and XGBoost is non-trivial, which limits the practical usage of the method.
4. Section 5 of the paper discusses the numerical instability of Vandermonde matrices in Linear TreeSHAP and uses the complex root of unity but does not cite prior work. In particular, the ill-conditioning problem of Vandermonde matrices and a similar solution for Linear TreeSHAP was previously mentioned in [1].

[1] Jiang, Z., Zhang, M., & Zhang, D. (2025). Fast Calculation of Feature Contributions in Boosting Trees. Proceedings of the 41st Conference on Uncertainty in Artificial Intelligence (UAI).

**Questions:**

How to set T and $\epsilon$ in practice?

---

> ### Author Response · Authors · 2025-11-25
>
> We appreciate the reviewer’s time and effort in providing constructive feedback. We have summarized our updates in our global reponse and address the concerns below.
>
> Q1: The abstract introduces several newly proposed names (TreeGrad, TreeGrad-Ranker, TreeProb and TreeGrad-Shap), which makes it difficult to identify the central contribution. It would be clearer and more engaging if the abstract can focus more on the main problem the paper aims to solve.
>
> A1: Thank you for your suggestion. We have restructured the abstract to better highlight our main contributions.
>
> Q2: It’s very difficult to distinguish the performance of different methods in Figure 3 based on solely color. I recommend the authors use additional visual cues, such as distinct shapes, line types for different groups of variables, or just numerical values to demonstrate the proposed methods.
>
> A2: Thank you for this suggestion. We have tried to alleviate this issue by only reporting the Beta Shapley value the either achieves the best (i) $\mathrm{Ins}(\pi; f_{\mathbf{x}})$, (ii) $\mathrm{Del}(\pi; f_{\mathbf{x}})$, or (iii) $\mathrm{Ins}(\pi; f_{\mathbf{x}})-\mathrm{Del}(\pi; f_{\mathbf{x}})$.
>
>
> Q3: Due to the lack of linearity for TreeGrad-Ranker, generalization to popular ensemble trees methods like random forests and XGBoost is non-trivial, which limits the practical usage of the method.
>
> A3: We wish to clarify:
> - Our Algorithm 2 also works for ensemble trees. After running Algorithm 2, it generates a trajectory $\mathcal{J}(f\_{\mathbf{x}}) = \\{\mathbf{z}\_{0},\mathbf{z}\_{1},\dots,\mathbf{z}\_{T-1}\\}$, and then $\zeta\_{f\_{\mathbf{x}}} = \sum\_{t=0}\^{T-1}\nabla \overline{f}\_{\mathbf{x}}(\mathbf{z}\_{t})$. The nonlinearity comes from that $\mathcal{J}(f\_{\mathbf{x}})$ is nonlinear in $f\_{\mathbf{x}}$. Nevertheless, $\overline{f}\_{\mathbf{x}}$ is still linear in $f\_{\mathbf{x}}$.
>
> - To illustrate our point, we have additionally used GradientBoostingClassifier and GradientBoostingRegressor from the scikit-learn library to train gradient boosted trees, and compared our methods with the considered baselines. These results can be found in Figures 9-13.
>
>
> Q4: Section 5 of the paper discusses the numerical instability of Vandermonde matrices in Linear TreeSHAP and uses the complex root of unity but does not cite prior work. In particular, the ill-conditioning problem of Vandermonde matrices and a similar solution for Linear TreeSHAP was previously mentioned in [1].
>
> A4: Thank you for sharing this highly relevant paper, which we have cited and discussed in our revision (lines 420–423). We note that using complex roots does not yield a fully numerically stable algorithm compared to our TreeGrad-Shap derived from the multilinear extension. As shown in Figure 2, the numerical error of using complex roots can be up to $10\^{6}$ times larger than that of our TreeGrad-Shap.
>
>
> Q5: How to set $T$ and $\epsilon$ in practice?
>
> A5: Based on our experience, these parameters can be set as follows:
> - We have additionally added Algorithm 9 to show that $\overline{f}(\mathbf{z})$ can be computed in $O(L)$ time. It is derived from Algorithm 2 by removing the traversing-up procedure. In our experiment, we selected the largest $\epsilon$ in the candidate set $\\{\dots,0.1, 0.5, 1, 5, 10, \dots\\}$ that provides consistent convergence in maximizing $\cfrac{1}{2}(\overline{f}\_{\mathbf{x}}(\mathbf{z})-\overline{f}\_{\mathbf{x}}(\mathbf{1}\_{N}-\mathbf{z}))$, which gives $\epsilon=5$.
>
> - We included additional results in Figures 14-23, showing the performance for different combinations of $T$ and $\epsilon$. Our observations are: (i) the resulting performance improves and converges as $T$ increases; (ii) TreeGrad-Ranker with ADAM converges earlier than gradient ascent, as performance with ADAM reaches near convergence across all the used datasets at $T=10$; and (iii) lower $\epsilon$ does not improve performance.
>
> We have added this discussion to Appendix G.

---

> > ### Comment · Reviewer_WPUD · 2025-11-26
> >
> > Thank you for the modifications and explanations.
> >
> > The presentation of the figures is much clearer, and I appreciate the clarification on the extension to GBDT as well as the settings of T and $\epsilon$. I have raised my score accordingly.

---

### Official Review · Reviewer_Kxc7 · 2025-10-31

**Soundness:** 3
**Presentation:** 3
**Contribution:** 3
**Rating:** 8
**Confidence:** 3

**Summary:**

This work focuses on constructing a feature importance ranking for tree models. The authors concentrate on building rankings based on deletion and insertion scores. While such rankings can be constructed naively in theory, this approach is NP-hard. Traditionally, probabilistic values have been used as scores for constructing rankings, including the popular Shapley values and the less popular Banzhaf values. The authors provide a theoretical explanation of why probabilistic values are not appropriate for this task. They then propose their own approach to constructing scores for ranking by formulating a new objective that should yield better ranking scores. They demonstrate how calculating gradients can be utilized in this framework. The authors show that their gradient methods can compute any probabilistic value, including popular Shapley values and improve upon existing methods. They implemented Shapley values calculation as a benchmark and compare their approach to existing methods, demonstrating improvements and explaining issues with previous approaches. The paper concludes with an empirical evaluation of ranging generated with TreeGrad-Ranker.

**Strengths:**

* The authors clearly motivated the problem through Proposition 1, where they stated a type of impossibility theorem for probabilistic values.
In Proposition 2, they showed that the use of probabilistic values is equivalent to substituting linear surrogates for f(x), which provides a novel perspective on building rankings with probabilistic values.

* They clearly stated that optimizing the proposed score can be formulated as gradient optimization and showed the exact form of this gradient.
* The authors demonstrated an interesting connection between gradient calculation and calculating probabilistic values, introducing a new way of computing them (including Shapley values, which is an active and important research topic).

* They identified problems in previous approaches and designed and conducted experiments to showcase these issues.

* They evaluated the new ranking on multiple datasets with a varying number of features.

**Weaknesses:**

* *Proposition 1* - In this proposition, you assume that U is any function 2^N → R, which is of course true in terms of game theory. However, here we are in a use case where we have a model that takes all features and we calculate expected values (based on the conditional distribution given by the proportion in the tree). It seems that the tree plus the distribution that is also defined by the tree may be less expressive. In Bilodeau et al. (2024), the only requirement for the model class was to have 3 piecewise linear regions and non-zero probability outside of a neighborhood (though this was for marginal SHAP). Did you think about stating this theorem in the context of tree models?

* Do I understand correctly that the proposed algorithm applies only to single decision trees? If so, this limits the scope of the paper, as gradient boosted trees are widely adopted and used very often. Single trees are more interpretable, but that is precisely what makes a feature ranking method for forests needed. Can it be extended to models containing multiple trees?

* In the empirical section (Section 6), the hyperparameters of the model are not described. Only the implementation used is described. However, there is no information about the hyperparameter selection process - I think it would be important to ensure that these trees are reasonably well-tuned for the datasets. Also, the size of the trained models was not reported. (See also questions).

* The authors evaluated their method using the insertion and deletion scores. There are also other methods for benchmarking the quality of rankings, e.g., Agarwal, Chirag, et al. "OpenXAI: Towards a transparent evaluation of model explanations." The theoretical justification aligns with the fact that it works better for these scores; however, what about other metrics?

**Questions:**

* Regarding the use of gradient optimization and the specific step in Line 23 of Algorithm 2: Do you know if the optimization results typically converge to a global minimum or a local minimum? The landscape of this problem may differ from typical ML problems (like neural networks). Did the authors consider or experiment with alternative optimizers (e.g., SGD or ADAM)? For smaller datasets, is it possible to find the optimal solution to this problem and use it as a comparison baseline?

* Theoretical Constraints (Impossibility Theorem) Is the work subject to the constraints of the relevant Impossibility Theorem from Bilodeau et al. (2024) ? Specifically, does this theorem hold for the values calculated in Algorithm 2, or do the methods presented in the paper mitigate these theoretical problems?

* Scores as Feature Attribution - Did the authors consider using the scores generated by Algorithm 2 for feature attribution? These scores appear to fulfill the most important axioms for feature attribution. Have the authors explored whether they offer improved performance or desirable behavior in this context?

---

> ### Author Response · Authors · 2025-11-25
> **Response (1/3)**
>
> We sincerely thank the reviewer for the valuable comments. We have summarized our updates in our global reponse and address the concerns below.
>
> Q1: Proposition 1 - In this proposition, you assume that U is any function 2^N → R, which is of course true in terms of game theory. However, here we are in a use case where we have a model that takes all features and we calculate expected values (based on the conditional distribution given by the proportion in the tree). It seems that the tree plus the distribution that is also defined by the tree may be less expressive. In Bilodeau et al. (2024), the only requirement for the model class was to have 3 piecewise linear regions and non-zero probability outside of a neighborhood (though this was for marginal SHAP). Did you think about stating this theorem in the context of tree models?
>
> A1: The impossibility theory in Bilodeau et al. (2024) can be adjusted for decision trees, and we provide a sketched proof. Note that their hypothesis aims to distinguish two different beheviors of a trained model $f$ within the sufficiently small neighborhood $(\mathbf{x}-\delta, \mathbf{x}+\delta)$ around the explicand $\mathbf{x}$. For a trained decision tree $f$, its output within a sufficiently small neighborhood around $\mathbf{x}$ would be constantly $f(\mathbf{x})$. Therefore, the counterpart null hypothesis for decision trees becomes
> $$
> \mathcal{F}^{0} = \\{f\in\mathcal{F}\colon  f(\mathbf{x}') = c_{0}\\},
> $$
> whereas
> $$
> \mathcal{F}^{1} =\\{f\in\mathcal{F}\colon f(\mathbf{x}') = c_{1}\\}.
> $$
> Next, their assumption that $\mathcal{F}$ contains 3-piecewise linear functions can be replaced by the assumption that $\mathcal{F}$ contains decision trees splitting on a single feature. Therefore, we can construct a decision tree $f_{j}$ splitting only on the $j$-th feature by letting
> $$
> f\_{j}(x'\_{j}) = \begin{cases}
> L\_{j}, & x'\_{j} \leq x\_{j}-\delta\_{j},\\\\
> c\_{0}/p, & x\_{j}-\delta\_{j} < x'\_{j} \leq x\_{j}+\delta\_{j},\\\\
> R\_{j}, & x'\_{j} > x\_{j}+\delta\_{j}
> \end{cases}
> $$
> where $p$ is the number of features, and $L_{j}$ and $R_{j}$ are undetermined constants. Note that each of their 3-piecewise linear functions, denoted by $f'\_{j}$, also contain two undetermined constants $\beta\^{L}\_{j}$ and $\beta\^{R}\_{j}$. The key step in proving Theorem B.4 therein is that the assumption that the attribution method $\phi$ is complete offers an equation to determine the values of $\beta\^{L}\_{j}$ and $\beta\^{R}\_{j}$. In particular, $\phi\_{j}(f'\_{j})$ can be anything by controlling $\beta\^{L}\_{j}$ and $\beta\^{R}\_{j}$. Following the same assumption and argument, $L_{j}$ and $R_{j}$ in the above $f_{j}$ can also be used to control $\phi\_{j}({f\_{j}})$. In other words, $\phi(f)$ can be anything for $f \in \mathcal{F}\_{0}$. This is sufficient to extend their impossibility theory to decision trees.
>
>
> Q2: Do I understand correctly that the proposed algorithm applies only to single decision trees? If so, this limits the scope of the paper, as gradient boosted trees are widely adopted and used very often. Single trees are more interpretable, but that is precisely what makes a feature ranking method for forests needed. Can it be extended to models containing multiple trees?
>
>
> A2: We wish to clarify:
> - Our Algorithm also works for gradient boosted trees, because each gradient $\nabla\overline{f}\_{\mathbf{x}}(\mathbf{z})$ is still linear in $f\_{\mathbf{x}}$, which can be verified using Eq. (7). The nonlinearity comes from that the trajecotry of $\mathbf{z}$ obtained while runing gradient ascent is nonlinear in $f\_{\mathbf{x}}$.
> - We have additionally used GradientBoostingClassifier and GradientBoostingRegressor from the scikit-learn library to train gradient boosted trees, and compared our methods with the considered baselines. These results can be found in Figures 9-13.
>
>
> Q3: In the empirical section (Section 6), the hyperparameters of the model are not described. Only the implementation used is described. However, there is no information about the hyperparameter selection process - I think it would be important to ensure that these trees are reasonably well-tuned for the datasets. Also, the size of the trained models was not reported. (See also questions).
>
> A3: Thank you for these questions and suggestions.
> - We only set the tree depths to those reported in the figures and the random state of training is set to $2025$ for reproducibility. All other hyperparameters are kept at their default values. For the added gradient-boosted trees, due to time constraints, we simply set the number of trees to $5$.
> - We have added Table 2 to report the sizes of the trained models and thier test performance.

---

> ### Author Response · Authors · 2025-11-25
> **Response (2/3)**
>
> Q4: The authors evaluated their method using the insertion and deletion scores. There are also other methods for benchmarking the quality of rankings, e.g., Agarwal, Chirag, et al. "OpenXAI: Towards a transparent evaluation of model explanations." The theoretical justification aligns with the fact that it works better for these scores; however, what about other metrics?
>
> A4:
>
> - Some faithfulness metrics therein rely on the existence of ground truth explanations. The provided use case is logistic regression, $f(\mathbf{x}) = \sigma(\mathbf{w}\cdot\mathbf{x})$, where the coefficient vecotr $\mathbf{w}$ is employed as the ground truth explanation. However, to our knowledge, there are no commonly-used ground truth explanations for decision trees. Therefore, these metrics are not applicable in our context.
>
> - For the other faithfulness metrics PGI and PGU that do not require ground trouth explanations, we found that they could be constantly zero for decision trees when the used neighboorhood is sufficiently small. Precisely, let $\zeta$ be an explanation for $f(\mathbf{x})$, the top-$k$ PGI is defined as
> $$
> \mathbb{E}\_{\mathbf{x}'}[|f(\mathbf{x}) - f(\mathbf{x}')|]
> $$
> where $x'\_{i} = x\_{i} + 1\{i \in \mathrm{top}\_{k}(\zeta)\}\cdot z\_i$ with $z_i \sim \mathcal{N}(0,\sigma)$ and $\mathrm{top}_{k}(\zeta)$ returns the $k$ largest elements of $|\zeta|$. If $\sigma$ is sufficiently small, the resulting top-$k$ PGI would be $0$. This is because the output of the decition tree $f$ would be constantly $f(\mathbf{x})$ within the sufficiently small neighborhood around $\mathbf{x}$. This is also true for PGU.
>
> - The stability metircs RIS and ROS contain the scalar $\|\zeta_{f_{\mathbf{x}}} - \zeta_{f_{\mathbf{x}'}}\|$ where $\zeta_{f_{\mathbf{x}}}$ denotes the vector of feature scores returned by any of the attribution methods in our paper and $\mathbf{x}'$ is within the neighborhood around $\mathbf{x}$. In particular, $f_{\mathbf{x}'}$, as defined in Algorithm 3, could also be constant within a sufficiently small neighborhood around $\mathbf{x}$. Consequently, $\|\zeta_{f_{\mathbf{x}}} - \zeta_{f_{\mathbf{x}'}}\| = 0$. In other words, RIS and ROS could be constantly zero in our context.
>
>
> Q5: Regarding the use of gradient optimization and the specific step in Line 23 of Algorithm 2: Do you know if the optimization results typically converge to a global minimum or a local minimum? The landscape of this problem may differ from typical ML problems (like neural networks). Did the authors consider or experiment with alternative optimizers (e.g., SGD or ADAM)? For smaller datasets, is it possible to find the optimal solution to this problem and use it as a comparison baseline?
>
> A5: Thank you for these questions.
> - To our knowledge, there is no global convergence guarantee even if the objective in Eq. (5) is assumed to be submodular.
> - It is certain that Algorithm 3 could converge to local maxima if the initialized $\mathbf{z}$ is close to them. Specifically, a local maximum $S$ is characterized by (i) $U(S\cup\{i\}) \leq U(S)$ for every $i \not\in S$ and (ii) $U(S\setminus\{i\}) \leq U(S)$ for every $i \in S$. This motivates us to initialize $\mathbf{z}$ as $0.5\cdot \mathbf{1}$, so that it is equally distant from all potential local maxima.
> - We have added Algorithm 8 to show how to run our TreeGrad-Ranker with ADAM, and included it in all figures for comparison. In particular, the resulting feature score vector is consistent with Theorem 1, which we have updated accordingly.
>
> - If the ground truth ranking is defined to be the optimal permutation that maximizes $\mathrm{Ins}(\pi; f_{\mathbf{x}}) - \mathrm{Del}(\pi; f_{\mathbf{x}})$, then a brute-force search would require $\Theta(N!)$ time. A more efficient approach exists via backward dynamic programming (see [1]), which reduces the complexity to $\Theta(2^{N})$ instead. However, this remains intractable for the employed datasets, where the smallest number of features is $32$.
>
>
>
>
> **References**
>
> [1] Chi, H., Wu, Q., Zhou, Z., Light, J., Dodwell, E., & Ma, Y. (2025). Unifying and Optimizing Data Values for Selection via Sequential-Decision-Making. arXiv preprint arXiv:2502.04554.

---

> ### Author Response · Authors · 2025-11-25
> **Response (3/3)**
>
> Q6: Theoretical Constraints (Impossibility Theorem) Is the work subject to the constraints of the relevant Impossibility Theorem from Bilodeau et al. (2024) ? Specifically, does this theorem hold for the values calculated in Algorithm 2, or do the methods presented in the paper mitigate these theoretical problems?
>
> A6: We wish to clarify:
> - The impossibility theorem in that work relies on the assumptions that the attribution method is both linear and complete. In contrast, our Proposition 1 only assumes that $\phi(f_{\mathbf{x}})$ is linear in $f_{\mathbf{x}}$, which we have clarified in the revision. In particular, their notion of linearity differs from ours. Their notion of linearity states that if $f(\mathbf{x}) = \sum_{i \in [N]}f^{(j)}(x_j)$, then $\phi_{j}(f, \mathbf{x}) = \phi(f^{(j)}, x_{j})$.
>
> - For our Algorithms 2 and 8, the produced vector of feature scores $\zeta$ is not linear in $f_{\mathbf{x}}$. Therefore, the proof underlying Proposition 1 does not apply to our TreeGrad-Ranker.
>
>
>
> Q7: Scores as Feature Attribution - Did the authors consider using the scores generated by Algorithm 2 for feature attribution? These scores appear to fulfill the most important axioms for feature attribution. Have the authors explored whether they offer improved performance or desirable behavior in this context?
>
> A7: To our knowledge, there is no commonly-used metric that directly assesses feature scores rather than the rankings they induce. Because it is unclear how to effectively and reliably evaluate feature scores, we choose to evaluate rankings in our experiments.

---

### Official Review · Reviewer_SCwW · 2025-11-02

**Soundness:** 2
**Presentation:** 2
**Contribution:** 3
**Rating:** 4
**Confidence:** 3

**Summary:**

This paper deals with feature ranking of decision trees via probabilistic values, such as the well-known Shapley values. The quality of feature ranking is assessed with the insertion and deletion metrics. Empirically, optimizing these two metrics is closely related to optimizing a particular objective. The authors find that solving this optimization leads to generally unreliable probabilistic values.

To address it, this paper optimizes the multilinear extension of the objective via gradient ascent called TreeGrad. Then, the averaged gradient is used to estimate the probabilistic values of features. The proposed method is better on insertion and deletion metrics.

**Strengths:**

1. The effectiveness of proposed method is supported by experimental results. That is both insertion and deletion metrics are significantly better than baselines.
2. I appreciate the authors’ dedicated efforts on the problem of feature ranking of decision trees.

**Weaknesses:**

1. The representation should be improved.

a) Too many abbreviation in abstract section (TreeGrad/TreeGrad-Ranker/TreeProb/TreeGrad-Shap/Linear TreeShap). Some of them are presented without further elaboration.

b) The logic flow of introduction section is unclear. It arises a question: how reliable are probabilistic values in ranking features? However, it is unclear that whether the probabilistic values itself are unreliable or the optimization method is unreliable. If it were the former, the paper would fall apart.

c) The contribution summary part in introduction section lacks a clear focus. The core contribution is TreeGrad. While TreeProb/TreeGrad-Shap/TreeGrad-Ranker are byproduct contributions. Thus, the author should highlight the development of TreeGrad.

d) The overall representation lacks the description of key ideas and motivations. The current version is hard to follow and it should be restructured.

2. The key contribution of this paper is using the multi-linear extension of objective and develop an algorithm to compute its gradient. However, it is unclear why the multi-linear extension is better. Is it possible to directly compute the gradient of original objective?

**Questions:**

-

---

> ### Author Response · Authors · 2025-11-25
>
> We appreciate your thoughtful and constructive review. We have summarized our updates in our global reponse and address the concerns below.
>
> Q1: Too many abbreviation in abstract section (TreeGrad/TreeGrad-Ranker/TreeProb/TreeGrad-Shap/Linear TreeShap). Some of them are presented without further elaboration.
>
> A1: Thank you for your suggestion. We have restructured the abstract to better highlight our main contributions in the revision.
>
>
> Q2: The contribution summary part in introduction section lacks a clear focus. The core contribution is TreeGrad. While TreeProb/TreeGrad-Shap/TreeGrad-Ranker are byproduct contributions. Thus, the author should highlight the development of TreeGrad. The overall representation lacks the description of key ideas and motivations. The current version is hard to follow and it should be restructured.
>
> A2: Thank you for your suggestion. In the revision, we have restructured our contributions in the introduction to better present our key ideas and motivations (lines 73-100) and highlight TreeGrad as the backbone.
>
>
>
> Q3: The logic flow of introduction section is unclear. It arises a question: how reliable are probabilistic values in ranking features? However, it is unclear that whether the probabilistic values itself are unreliable or the optimization method is unreliable. If it were the former, the paper would fall apart.
>
> A3: We wish to clarify:
> - If we understand correctly, the optimization method may refer to $\arg\min\_{g\in\mathcal{L}}\sum\_{S\subseteq [N]}\eta\_{s+1}(f\_{\mathbf{x}}(S)-g(S))\^{2}$ in Proposition 2, which is the only optimization related to probabilistic values. We would like to clarify that this proposition aims to convey that each probabilistic value $\phi$ can be cast as the linear surrogate, i.e., $g\^{\*}(S) = c\_{0} + \sum\_{i\in S}(\phi\_{i}+c)$, closest to $f\_{\mathbf{x}}$ in the least-square sense. If we substitute $g\^\*$ for $f\_{\mathbf{x}}$ in Eq. (5), this optimization can be easily solved using the coefficients of $g^*$, or equivalently $\phi$. This is how we connect probabilistic values to solving Eq. (5). Since solving Eq. (5) is closely related to jointly optimizing the insertion and deletion metrics, probabilistic values are therefore also connected to this joint optimization.
>
> - For decision trees, each probabilistic value can be accurately computed using Algorithm 4 and there is no need to run any optimization. Therefore, our established unreliability is for probabilistic values themselves.
>
> - In our revision, we clarified that the unreliability established in Proposition 1 relies only on the assumption that $\phi(f_{\mathbf{x}})$ is linear in $f_{\mathbf{x}}$, which includes all probabilistic values.
>
>
>
>
> Q4: The key contribution of this paper is using the multi-linear extension of objective and develop an algorithm to compute its gradient. However, it is unclear why the multi-linear extension is better. Is it possible to directly compute the gradient of original objective?
>
> A4: We wish to clarify:
> - Let $g(S)=\frac{1}{2}(f_{\mathbf{x}}(S)-f_{\mathbf{x}}([N]\setminus S)$ be the objective in Eq. (5), and observe that its domain is $\{S\subseteq N\}$. As far as we know, for such a discrete domain, we can only have discrete gradients defined as $\Delta_i g(S) = g(S\cup\{i\}) - g(S\setminus\{i\})$, which themselves are not clearly linked to maximizing $g(S)$.
> - The multilinear extension $\overline{g}(\mathbf{z})$ of $g(S)$ makes the domain continuous, allowing the (continuous) gradient $\nabla\overline{g}(\mathbf{z})$ to be defined, which in turn enables maximizing $\overline{g}(\mathbf{z})$ using gradient ascent. As proved in Proposition 3, maximizing $\overline{g}(\mathbf{z})$ is equivalent to maximizing $g(S)$, which justifies the use of $\overline{g}(\mathbf{z})$ in the context.
> - As added in Appendix B, any symmetric semi-value, which includes the Shapley and Banzhaf values, can be expressed as an expectation of the gradient field of $\overline{g}(\mathbf{z})=\frac{1}{2}(\overline{f}\_{\mathbf{x}}(\mathbf{z})-\overline{f}\_{\mathbf{x}}(\mathbf{1}\_{N}-\mathbf{z}))$. This motivates us to average the gradients while maximizing $\frac{1}{2}(\overline{f}\_{\mathbf{x}}(\mathbf{z})-\overline{f}\_{\mathbf{x}}(\mathbf{1}\_{N}-\mathbf{z}))$ as a vector of feature scores, leading to our TreeGrad-Ranker.  In particular, as proved in our Theorem 1, our TreeGrad-Ranker derived from this gradient field possesses all the axioms uniquely characterizing probabilistic values, except for the linearity, which itself leads to the unreliability estabilished in our Proposition 1.

---

### Author Response · Authors · 2025-11-25
**Updates in the Revision**

We thank all reviewers for their valuable time and constructive comments. We summarize the main updates made in the revision. Changes are marked in red.

- We have restructured the contributions in the abstract and introduction to better present our key ideas and motivations (Reviewers SCwW, WPUD and vun4).
- We clarified Proposition 1 to emphasize that the unreliability result only relies on the assumption that $\phi(f_{\mathbf{x}})$ is linear in $f_{\mathbf{x}}$ (Reviewers SCwW and Kxc7).
- We further implemented our TreeGrad-Ranker using the ADAM optimizer, as presented in Algorithm 8. The produced feature scores are consistent with Theorem 1, which has been updated accordingly, and the results of TreeGrad-Ranker with ADAM are included (Reviewer Kxc7).
- To better distinguish the performance of different methods, we report only the Beta Shapley values that achieve the best result in one of the following: (i) $\mathrm{Ins}(\pi; f_{\mathbf{x}})$, (ii) $\mathrm{Del}(\pi; f_{\mathbf{x}})$, or (iii) $\mathrm{Ins}(\pi; f_{\mathbf{x}})-\mathrm{Del}(\pi; f_{\mathbf{x}})$, instead of showing all Beta Shapley values (Reviewers WPUD and vun4).
- In Appendix B, we elaborated on how any symmetric semi-value can be expressed as an expectation of $\nabla \cfrac{1}{2}(f_{\mathbf{x}}(\mathbf{z})-f_{\mathbf{x}}(\mathbf{1}_{N}-\mathbf{z}))$, which motivates the design of our TreeGrad-Ranker (Reviewer SCwW).
- Since (gradient boosted) decision trees could not fit the ONP dataset (39,644 instances and 59 features) well, we have replaced it with a larger BT dataset (583,250 instances and 77 features).
- We used GradientBoostingClassifier and GradientBoostingRegressor from the scikit-learn library to train gradient boosted trees, and the corresponding results are presented in Figures 9-13. This demonstrates that our TreeGrad-Ranker is not limited to decision trees (Reviewers Kxc7 and WPUD).
- We included Table 2 to summarize the sizes of the used tree models and their test performance (Reviewer Kxc7).
- In Appendix G, we added an experiment comparing the actual runtime of Linear TreeShap, TreeGrad-Shap and TreeGrad, with result shown in Figure 5 (Reviewer vun4).
- We also added guidance in Appendix G on selecting $\epsilon$ and $T$. Specifically, we conducted experiments comparing the performance across different combinations of $T$ and $\epsilon$, with results shown in Figures 14-23 (Reviewers WPUD and vun4).

For reproducibility, we will release our code once the submission is de-anonymized.

---

### Author Response · Authors · 2025-12-02
**Summary**

We sincerely thank all reviewers for their efforts and constructive feedback, and we also appreciate the AC’s efforts. Below, we summarize the changes made during the discussion phase.

As detailed in our earlier global response, we revised and improved the manuscript accordingly, with all modifications marked in red. The main changes include:
- restructuring our key ideas in the abstract and introduction to better present our contributions (Reviewers SCwW and WPUD);
- adding further experimental details and results in Appendix G and in the updated figures, along with improved figure presentation (Reviewers Kxc7, WPUD and vun4).

During the discussion, Reviewer WPUD raised the score from 4 to 6, as our response addressed his/her concerns.

Although we lost the opportunity to hear back from Reviewers SCwW, Kxc7 and vun4, we are confident that our responses have also sufficiently addressed their concerns.

---

### Meta-Review · Area_Chair_h4vJ · 2026-01-11

**Summary:**

The paper studies feature attribution for explaining decision tree predictions, showing that traditional probabilistic values such as Shapley and Banzhaf are unreliable for jointly optimizing insertion and deletion metrics. To address this, the Authors propose TreeGrad, an efficient gradient-based method for optimizing a joint objective, and introduce TreeGrad-Ranker for improved feature ranking and TreeGrad-Shap for numerically stable Shapley value computation. Experiments demonstrate superior ranking performance and reduced numerical errors compared to existing methods.

The initial scores are relatively widely spread, ranging from 4 to 8. The Reviewers raised several critical issues, including the presentation of the contribution, the focus on single decision trees, the lack of hyperparameter descriptions, evaluation limited to insertion and deletion metrics, absence of empirical runtime analysis, and experiments restricted to relatively small datasets. Overall, the Authors have addressed many of the Reviewers’ concerns, however, the revised manuscript appears to have undergone substantial changes. As such, it could benefit from another round of reviewing.

**Reviewer Concerns:**

Overall, the Authors have addressed the concerns raised in the Reviews. However, the revised manuscript appears to have undergone substantial changes. As such, it could benefit from another round of reviewing.

**Reviewer Scores:**

- Reviewer SCwW would likely increase their score from 4 to 5
- Reviewer Kxc7 could potentially decrease their score from 8 to 7 as the paper have undergone substantial changes
- Reviewer WPUD would likely increase their score from 4 to 5 or 6
- Reviewer vun4 would likely keep their score of 6

---

### Decision · Program_Chairs · 2026-01-26

Accept (Poster)